# TabFlowM: Lightweight flow matching for Mixed-Type Tabular Data Synthesis in Latent Space

**Yan Kin Chi**                                                    *yan_kin.chi@unsw.edu.au*
*School of Computer Science and Engineering*
*University of New South Wales*

**Andre Gunawan**                                                  *andre.gunawan@ntu.edu.sg*
*Digital Trust Centre (DTC) | NTU Singapore*
*Nanyang Technological University*

**Suryansh Maurya**                                                *sm0464@srmist.edu.in*
*Department of Networking and Communications*
*SRM Institute of Science and Technology*

**Harsh Kasyap**                                                   *hkasyap.cse@iitbhu.ac.in*
*Department of Computer Science and Engineering*
*Indian Institute of Technology (BHU) Varanasi*

**Raymond Wong**                                                   *ray.wong@unsw.edu.au*
*School of Computer Science and Engineering*
*University of New South Wales*

**Maple Carsten**                                                  *CM@warwick.ac.uk*
*University of Warwick*
*The Alan Turing Institute*

**Reviewed on OpenReview:** *https://openreview.net/forum?id=t5kygrpSIz*

## Abstract

Generative modeling for mixed-type tabular data has recently been dominated by diffusion-based methods, but their gains often come with schedule design, time dependent score parameterization, and multi-step solvers that increase computational overhead and tuning difficulty. We present **TabFlowM**, a lightweight framework that asks a more targeted question: once mixed-type records are mapped into a decoder compatible continuous transport space, is diffusion style score learning still necessary? TabFlowM answers this by training a single time conditioned velocity field via flow matching to deterministically transport Gaussian noise to the latent token space data distribution, replacing diffusion specific score estimation and scheduling machinery with direct velocity regression on a closed form coupling path. Experiments on six real world benchmark datasets show that TabFlowM attains the best average rank in composite distributional fidelity, jointly accounting for marginal and pairwise divergence. It further achieves the strongest column-wise MLE on 5 out of 6 datasets. Across the UCI suite, TabFlowM also trains in markedly less time than the strongest diffusion baselines, avoiding the severe training-time scaling they exhibit on larger datasets. Finally, on a million-scale fraud dataset with class ratios exceeding 100:1, where unconditional fidelity can decouple from rare event predictive utility, TabFlowM achieves the strongest average AUC-PR while maintaining competitive fidelity and runtime. These findings suggest that, under an appropriate transport interface for mixed-type data, a minimalist flow matching generator can recover much of the benefit commonly associated with heavier diffusion models while substantially reducing computational and conceptual complexity.

# 1 Introduction

Tabular data is among the most widely used data formats across domains such as finance Liu & Liu (2024), healthcare Nik et al. (2023), e-commerce Mendikowski & Hartwig (2022), and public policy Aghaddar et al. (2024). Synthetic tabular data has gained substantial attention as a means of enabling privacy-preserving data sharing, scalable experimentation, and remediation of class imbalance, yet faithfully reproducing real world tabular data remains challenging: mixed numerical categorical domains, sharp sparsity patterns, and long range cross column dependencies demand generative models that are both expressive and computationally efficient Zhou et al. (2023).

Recent progress is driven by score-based continuous time diffusion models Kotelnikov et al. (2023); Shi et al. (2024); Zhang et al. (2024); Lee et al. (2023); Kim et al. (2023). Specifically, the class of elucidated diffusion models (EDM) Karras et al. (2022) adapt elegantly to mixed-type tables and have achieved SOTA performance in both fidelity and downstream predictive utility. A representative model *TabSyn* Zhang et al. (2024) maps numerical and categorical variables into a unified VAE latent space, allowing a single continuous-time model with a simple schedule and compact network to deliver strong fidelity at modest sampling cost. Building on this, *TabDiff* Shi et al. (2024) adds adaptively learned per-feature schedules, masked/absorbing categorical diffusion, and stochastic backward sampling, improving per-column consistency but at the expense of architectural and computational complexity Shi et al. (2024); Guzmán-Cordero et al. (2025).

Despite their popularity, EDM-style diffusion inherits several structural disadvantages. First, the training score target $\nabla_x \log p_t(x)$ changes scale dramatically across noise levels: as noise level $\sigma \to 0$, targets can explode, while for large $\sigma$ they vanish, forcing delicate, noise dependent preconditioning Lipman et al. (2023). Second, *high-fidelity diffusion sampling* often relies on careful numerical choices (e.g., solver settings and sampler heuristics) and many model evaluations, introducing extra hyperparameters and increasing inference and tuning cost Shi et al. (2024). Third, there is a training–inference mismatch: models are optimized via stochastic denoising objectives but deployed through a deterministic probability-flow ODE, so discretization and conditioning gaps can bias endpoints. Fourth, the noise schedule $\sigma(t)$ is fixed *a priori*; adapting it per feature or to data geometry effectively alters $p_t$ and requires re-deriving loss weights. Finally, in mixed-type tables, even when operating in a continuous latent space, calibrating discrete columns is only indirect, which can under-serve rare categories Gat et al. (2024); Lipman et al. (2023).

This work revisits a fundamental design question: is diffusion style score learning necessary when operating in a shared latent space for mixed-type tabular synthesis? We argue that much of the modeling burden attributed to diffusion can instead be handled by a simpler transport based formulation.

Subsequently we introduce **TabFlowM**: a mixed-type tabular synthesis framework that replaces EDM-style score learning with a pure *Flow Matching* (FM) objective. Motivated by the practical drawbacks of EDM, FM regresses a time dependent velocity field that directly transports a base distribution to the data distribution, offering three key advantages: (i) it targets well behaved velocities, alleviating noise-scale pathologies and reducing reliance on fragile preconditioning; (ii) it aligns training and sampling, since the same learned field is deterministically integrated at inference, narrowing the objective and sampler gap and often yielding more efficient training with competitive sampling cost; and (iii) it decouples learning from any fixed forward stochastic differential equation (SDE), enabling flexible transport paths without rederiving loss weights Lipman et al. (2023); Gat et al. (2024). Our contributions are as the following:

- **A controlled flow matching framework for mixed-type tabular synthesis**: We propose TabFlowM, which studies whether a single flow matched velocity field can replace diffusion style score learning once synthesis is carried out in a decoder compatible latent space, without auxiliary latent hierarchies, likelihood bounds, or moment-matching surrogates.

- **A transport interface for mixed-type data**: We show that mapping mixed-type records into a decoder compatible continuous token space makes lightweight Euclidean transport viable, and prove that in this space small pathwise velocity regression error yields endpoint control and Wasserstein convergence of the generated token-space distribution to the target distribution.

- **Extensive empirical comparison against diffusion baselines**: We evaluate TabFlowM on six public benchmarks and six million-scale fraud datasets with extreme class imbalance, showing that it delivers competitive or SOTA performance in fidelity, downstream predictive utility, privacy preservation, and runtime.

## 2 Problem Statement

We consider a mixed-type tabular dataset $\mathcal{T} = \{(x^{\text{num}}, x^{\text{cat}})\}$, where each data point consists of $d_{\text{num}}$ numerical features and $d_{\text{cat}}$ categorical features whose total dimension is $d = d_{\text{num}} + d_{\text{cat}}$. The numerical features are represented as a vector $x^{\text{num}} \in \mathbb{R}^{d_{\text{num}}}$, and the categorical features as a tuple $x^{\text{cat}} = (x_1^{\text{cat}}, \ldots, x_{d_{\text{cat}}}^{\text{cat}})$, where each $x_j^{\text{cat}} \in \{0, 1, \ldots, C_j - 1\}$ is a discrete category label drawn from a finite set of size $C_j$.

Our goal is to learn a generative model that can produce synthetic samples $\tilde{x} = (\tilde{x}^{\text{num}}, \tilde{x}^{\text{cat}})$ that are statistically similar to the true data distribution.

## 3 Related Work

The field of tabular data synthesis has evolved rapidly, with early approaches adapting deep generative models such as GANs and VAEs to handle heterogeneous feature types leading to pioneering model such as CTGAN and TVAE Xu et al. (2019). CTGAN introduced conditional generation based on frequency-sampled columns and applied mode-specific normalization to stabilize GAN training and better capture rare categories and multimodal distributions. TVAE followed a variational autoencoding framework, outputting each feature via a Gaussian or categorical head, enabling more faithful modeling of mixed-type marginals. While both models improved per-feature fidelity, they struggled to capture complex inter-feature dependencies.

To address this, transformer-based models treated rows as sequences to better model feature correlations. REaLTabFormer Solatorio & Dupriez (2023) employed an autoregressive GPT-style transformer to synthesize data, one feature at a time, capturing high-order dependencies and supporting relational synthesis. TabMT Gulati & Roysdon (2023) instead used a BERT-style masked transformer, enabling bidirectional feature modeling without relying on a fixed column order and naturally handling missing data. More recently, diffusion-based methods have emerged as the dominant paradigm for high-fidelity synthesis. STaSy Kim et al. (2023) introduced score-based diffusion with a self-paced training curriculum, improving diversity and stability. TabDDPM Kotelnikov et al. (2023) extended diffusion to mixed data by combining Gaussian and discrete forward processes. CoDi Lee et al. (2023) further refined this with dual cross-conditioned diffusion models for continuous and categorical features, improving dependency modeling across modalities. TabDDPM Kotelnikov et al. (2023) extended diffusion to mixed data by combining Gaussian and discrete forward processes.

TabSyn Zhang et al. (2024) proposed learning a score-based diffusion model in the latent space of a transformer-based VAE that tokenizes mixed-type columns into a continuous embedding; diffusion operates over this unified space and samples are decoded back to the original schema, yielding faster generation. The latest advance, TabDiff Shi et al. (2024), unifies all feature types within a single continuous-time diffusion process using feature-adaptive noise schedules and a transformer denoiser, achieving state-of-the-art results in both marginal fidelity and joint consistency.

Beyond standard diffusion pipelines, recent work has also explored ways to reduce the iterative cost of score-based generation. Score Mismatching (SMM) trains a standalone generator to compress the diffusion sampling trajectory into a single forward pass, while using a score network that both matches the real distribution and mismatches generated samples to provide informative gradients Ye & Liu (2024). Although SMM targets acceleration rather than tabular synthesis specifically, it reinforces the broader motivation that score-based models can be computationally burdensome and that alternative transport or generator based formulations may offer more efficient sampling.

Complementarily, Fisher Flow developed a geometry aware flow matching framework for discrete data by treating categorical distributions as points on a Fisher–Rao statistical manifold and mapping them to the positive orthant of a hypersphere, where geodesic paths and Riemannian optimal transport couplings can be

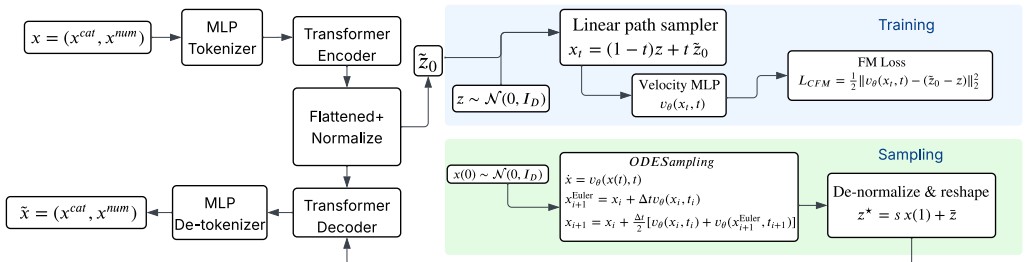

Figure 1: TabFlowM Framework

defined Davis et al. (2024). This line of work is closely aligned with our motivation to replace diffusion with flow based transport, but differs from TabFlowM in that it constructs a specialised manifold geometry for discrete variables, whereas we study whether a lightweight Euclidean FM objective becomes sufficient after mapping mixed-type tables into a decoder-compatible continuous token space.

Most recently, approaches such as Guzmán-Cordero et al. (2025); Jolicoeur-Martineau et al. (2024) explore utilizing FM as an auxiliary addition to more heavy designs such as explicit exponential-family parameterizations, variational moment matching, or per-time-step ensembles, leaving the benefits of a lean FM-based architecture underexplored.

## 4 Method

TabFlowM is organized around a simple principle: for mixed type tables, the generator should not be asked to solve heterogeneous representation and transport simultaneously. Accordingly, the two-stage pipeline in Fig. 1 first maps each record into a decoder compatible continuous token representation and normalizes it into the space where transport will be learned. With this interface fixed, the second stage keeps the generator deliberately minimal: a continuous-time flow matching model transports Gaussian noise to the token-space data distribution, after which the resulting samples are decoded back into tabular records.

The motivation is geometric. Flow matching is most natural when transport is performed in a continuous, approximately Euclidean space whose local neighborhoods are semantically meaningful Chen & Lipman (2024); Lipman et al. (2023); Gat et al. (2024). Raw mixed-type tables violate this premise: numerical columns may occupy incompatible scales, whereas categorical columns are discrete and induce discontinuous changes under naive Euclidean interpolation. Consequently, fitting a lightweight Euclidean velocity field directly in raw table space is unnecessarily difficult, because the model must handle heterogeneous feature geometry at the same time as it learns transport. TabFlowM therefore first constructs a decoder compatible transport space in which these irregularities are reduced, and only then applies a simple flow matching generator in that space.

### 4.1 Constructing a decoder compatible Transport Space

Given a mixed-type record $(x^{\mathrm{num}}, x^{\mathrm{cat}})$, the goal of the first stage is to construct a continuous representation that supports both accurate decoding and stable downstream transport. In particular, we seek a latent space in which all columns admit a common parameterization, nearby latent points decode to nearby records, and the encoded data distribution is sufficiently regular where lightweight continuous transport becomes well posed.

Let $d_{\mathrm{num}}$ and $d_{\mathrm{cat}}$ denote the numbers of numerical and categorical columns, respectively, and let $M := d_{\mathrm{num}} + d_{\mathrm{cat}}$ be the total number of columns. Let $d_e \in \mathbb{N}$ denote the token dimension. Categorical column $j$ has $C_j$ categories. Writing each categorical value $x_j^{\mathrm{cat}} \in \{0, \ldots, C_j - 1\}$ in one-hot form as $x_j^{\mathrm{oh}} \in \{0,1\}^{C_j}$, a record can be represented as

$$x \;=\; \left[x^{\mathrm{num}}, x_1^{\mathrm{oh}}, \ldots, x_{d_{\mathrm{cat}}}^{\mathrm{oh}}\right] \;\in\; \mathbb{R}^{d_{\mathrm{num}} + \sum_{j=1}^{d_{\mathrm{cat}}} C_j}.$$

**Why transport is performed in latent space.** As is, straight Euclidean interpolation is poorly aligned with mixed-type tabular structure. If $x$ and $x'$ differ only in one categorical column, then their one-hot encodings satisfy $\|x_j^{\text{oh}} - x'^{\text{oh}}_j\|_2 = \sqrt{2}$ regardless of whether the two categories are semantically similar or unrelated. Moreover, for any $t \in (0,1)$, the convex combination $(1-t)x_j^{\text{oh}} + tx'^{\text{oh}}_j$ lies in the interior of the simplex rather than at a valid vertex, so the straight segment immediately leaves the discrete data manifold. At the same time, heterogeneous numerical columns can induce anisotropic scaling, causing Euclidean distances to be dominated by only a few features. As a result, straight paths in raw table space are generally poor transport trajectories.

TabFlowM therefore performs transport in a learned latent space. The goal is not to make straight interpolation exact, but to move generation into a continuous representation where such paths are substantially better aligned with the underlying geometry. We call such a representation *decoder compatible* if each column can be accurately reconstructed from its own token alone, through a simple per-column linear head, without requiring cross-column nonlinear processing. This property is formalized in Definition 1 after all notation has been introduced. For now, we propose Proposition 1, with details shown in Appendix A.4.1:

**Proposition 1 (Raw-space mismatch versus decoder compatible token-space regularity)** *For any $t \in (0,1)$, the convex combination of two distinct one-hot categorical vectors is not a valid categorical state, so any straight transport path in raw mixed-type space leaves the discrete data manifold and forces the generative model to learn discontinuous corrections. In contrast, if the reconstruction map is twice continuously differentiable with uniformly bounded Hessian and strictly positive minimum Jacobian singular value, then a straight segment in decoder compatible token space reconstructs to a data-space trajectory whose normalized bending is bounded by a constant that depends only on the decoder's smoothness and conditioning, not on the choice of endpoints (Appendix A.4.1).*

The practical consequence for flow matching is direct: because straight token-space paths reconstruct to nearly straight data-space trajectories, a simple time-conditioned MLP can accurately regress the target velocity without needing to compensate for path curvature. This is the geometric reason a minimal generator suffices in this space but not in raw table space.

The regularity conditions invoked here, namely a uniformly bounded Hessian and a strictly positive minimum Jacobian singular value, are mild rather than restrictive. They hold generically for the smooth, well-conditioned decoders we employ, and can be threatened only in degenerate corner cases such as a categorical column collapsing onto a single vertex of its probability simplex; Appendix A.4.1 gives a per-head analysis of when the conditions hold, the sole regime in which they can fail, and empirical confirmation that no such degeneracy arises across our evaluation settings.

**Column Tokenizer.** We begin latentization by mapping each column into a shared continuous embedding space. Let the columns be indexed by $m \in [M]$, with $\mathcal{N} \subseteq [M]$ the numerical indices and $\mathcal{C} \subseteq [M]$ the categorical indices. Each column value $x_m$ is mapped to a token $t_m \in \mathbb{R}^{d_e}$ via

$$t_m = \begin{cases} x_m\, a_m + c_m, & m \in \mathcal{N}, \\ U_m[x_m] + c_m, & m \in \mathcal{C}, \end{cases} \tag{1}$$

where $a_m, c_m \in \mathbb{R}^{d_e}$ are learnable for numerical columns, and for a categorical column $m \in \mathcal{C}$ with $C_m$ categories, $U_m \in \mathbb{R}^{C_m \times d_e}$ is its embedding table and $U_m[x_m]$ denotes the row indexed by the category id $x_m$. Stacking the tokens yields:

$$E = \text{Stack}(t_1, \ldots, t_M) \in \mathbb{R}^{M \times d_e}. \tag{2}$$

This provides a common continuous parameterization across columns.

**Transformer Encoder (VAE).** To obtain a more regular latent representation, we encode the token sequence with a Transformer VAE. A transformer encoder $\text{Enc}_\phi$ maps $E$ to the parameters of a Gaussian latent distribution:

$$(\mu, \log \sigma^2) = \text{Enc}_\phi(E), \qquad z = \mu + \sigma \odot \xi, \ \xi \sim \mathcal{N}(0, I_\ell), \tag{3}$$

where $z \in \mathbb{R}^\ell$ and $\odot$ denotes elementwise multiplication. A transformer decoder then reconstructs the token matrix via

$$\widehat{E} = \mathrm{Dec}_\psi(z) \in \mathbb{R}^{M \times d_e}. \tag{4}$$

Note $z$ is not itself the operating space of the downstream flow model. Rather, it regularizes the autoencoder and induces a decoder compatible token representation, whose flattened and normalized form is used by the flow model in Section 4.2. The reconstruction term preserves record-level information, while the KL regularizer discourages poorly conditioned latent coordinates by pulling the approximate posterior toward a centered Gaussian. For $q_\phi(z \mid E) = \mathcal{N}(\mu, \mathrm{diag}(\sigma^2))$, the KL term is

$$\mathrm{KL}\big(q_\phi(z \mid E) \,\|\, \mathcal{N}(0, I_\ell)\big) = \frac{1}{2} \sum_{r=1}^{\ell} \left( \mu_r^2 + \sigma_r^2 - \log \sigma_r^2 - 1 \right). \tag{5}$$

This suppresses large coordinate-wise shifts and extreme variance imbalance, encouraging the encoded data cloud to occupy a more regular region of $\mathbb{R}^\ell$. After flattening and empirical normalization of the decoder compatible token representation, transport is better conditioned than in raw mixed-type space.

**Transformer Decoder (Column-wise Reconstruction).** The reconstructed tokens are projected back to the data space with heads mirroring the tokenizer. For numerical column $i$ and categorical column $j$,

$$\widehat{x}_i^{\mathrm{num}} = \widehat{e}_i^{\mathrm{num}} \widehat{w}_i^{\mathrm{num}} + \widehat{b}_i^{\mathrm{num}}, \quad \widehat{p}_j^{\mathrm{cat}} = \mathrm{Softmax}(\widehat{e}_j^{\mathrm{cat}} \widehat{W}_j^{\mathrm{cat}} + \widehat{b}_j^{\mathrm{cat}}), \tag{6}$$

with $\widehat{w}_i^{\mathrm{num}} \in \mathbb{R}^{d_e \times 1}$, $\widehat{b}_i^{\mathrm{num}} \in \mathbb{R}$, $\widehat{W}_j^{\mathrm{cat}} \in \mathbb{R}^{d_e \times C_j}$, and $\widehat{b}_j^{\mathrm{cat}} \in \mathbb{R}^{1 \times C_j}$. The categorical reconstruction is a probability vector $\widehat{p}_j^{\mathrm{cat}} \in \Delta^{C_j - 1}$ over the categories of column $j$. We write $\widehat{x} = (\widehat{x}^{\mathrm{num}}, \widehat{x}^{\mathrm{cat}})$, with $\widehat{x}_j^{\mathrm{cat}}$ sampled from taking the argmax of $\widehat{p}_j^{\mathrm{cat}}$ at generation time. With the full reconstruction pipeline now defined, we formalize the definition:

**Definition 1 (Decoder-compatible token space)** *Let* $\mathrm{Dec}_\psi$ *denote the learned transformer decoder and let* $R_m$ *denote the per-column reconstruction head for column $m$ (Eq. 6). The continuous representation* $\widehat{E} = \mathrm{Dec}_\psi(z_{1:}) \in \mathbb{R}^{M \times d_e}$ *is* decoder compatible *if it satisfies:*

1. ***Column-local decodability.*** *Each reconstructed value* $\widehat{x}_m = R_m(\widehat{t}_m)$ *depends only on the token* $\widehat{t}_m$ *of column $m$, through an affine map (numerical columns) or a linear projection to category logits (categorical columns). Decoding does not couple tokens across columns.*

2. ***Reconstruction sufficiency.*** *Applying these column-local heads to* $\widehat{E}$ *achieves low reconstruction error under the training distribution. That is, column-local decoding is not merely well-defined but* sufficient *for accurate recovery.*

This distinguishes $\widehat{E}$ from the encoder output $\mu$: both have shape $M \times d_e$ and can be passed through the same linear heads, but only $\widehat{E}$ yields accurate reconstructions without the cross-column transformer decoder. At embedding extraction time, we set $z_{1:} = \mu_{1:}$ (i.e., the posterior mean without reparameterization noise), yielding a deterministic $\widehat{E}$ for downstream flow matching. Property 1 is enforced architecturally by the column-wise heads in Eq. equation 6. Property 2 is encouraged by the VAE objective (Eq. 7) and validated empirically. Throughout this paper, *decoder-compatible token space* refers to the flattened representation $z_0 = \mathrm{vec}(\widehat{E}) \in \mathbb{R}^D$ satisfying Definition 1.

**Objective with Adaptive Weight.** The VAE is trained with a composite reconstruction loss together with a KL penalty:

$$\mathcal{L}(\phi, \psi) = \ell_{\mathrm{recon}}(x, \widehat{x}) + \beta \, \mathrm{KL}\big(q_\phi(z \mid E) \,\|\, \mathcal{N}(0, I_\ell)\big), \tag{7}$$

where

$$\ell_{\mathrm{recon}}(x, \widehat{x}) = \sum_{i=1}^{d_{\mathrm{num}}} \lambda_i^{\mathrm{num}} \left\| x_i^{\mathrm{num}} - \widehat{x}_i^{\mathrm{num}} \right\|_2^2 - \sum_{j=1}^{d_{\mathrm{cat}}} \lambda_j^{\mathrm{cat}} \langle x_j^{\mathrm{oh}}, \log \widehat{p}_j^{\mathrm{cat}} \rangle. \tag{8}$$

The weights $\lambda_i^{\mathrm{num}}, \lambda_j^{\mathrm{cat}} > 0$ balance heterogeneous units and category prevalences across columns.

Because the downstream flow model operates on the flattened and normalized decoder compatible token representation rather than directly on $z$, we avoid over-regularizing the VAE latent space and employ an adaptive schedule that decreases $\beta$ once reconstruction has plateaued. Let $\bar{\ell}_m(t)$ be an exponential moving average of the contribution of column $m \in [M]$ to $\ell_{\text{recon}}$ at epoch $t$. For patience $S \in \mathbb{N}$ and tolerance $\tau > 0$, define

$$\Delta(t) \;=\; \frac{1}{M} \sum_{m=1}^{M} \big(\bar{\ell}_m(t) - \bar{\ell}_m(t-S)\big). \tag{9}$$

With initialization $\beta \leftarrow \beta_{\max}$, we update

$$\beta \;\leftarrow\; \begin{cases} \max\{\beta_{\min}, \lambda\beta\}, & \text{if } \Delta(t) \geq -\tau, \\ \beta, & \text{otherwise,} \end{cases} \tag{10}$$

where $\lambda \in (0,1)$ and $0 < \beta_{\min} \leq \beta_{\max}$. This schedule initially promotes a well-conditioned latent organization and later relaxes the KL pressure so that finer record-level structure is retained for downstream generation.

**Interface to TabFlowM.** We now enter the generative stage. By Proposition 1, straight paths in the decoder compatible token representation decode to trajectories with controlled bending, whereas straight paths in raw one-hot space leave the data manifold entirely. The VAE can therefore map records into a space where simple Euclidean paths are geometrically well behaved, so that a lightweight velocity field can follow them without compensating for path curvature. Accordingly, rather than learning a score field under an explicit forward noising process, TabFlowM adopts flow matching on the flattened and normalized decoder compatible token representation. Among admissible path constructions, we choose a linear coupling between Gaussian noise and encoded data because it is analytically closed form and implementation light. The learned object is therefore a time dependent velocity field whose induced ODE transports the base distribution to the learned token space data distribution.

## 4.2 Flow Matching Generator in decoder compatible Token Space

Having constructed a regularized decoder compatible token space, TabFlowM adopts a deliberately *minimal* continuous-time generator. Specifically, we flatten and normalize the token representation, choose a linear coupling between Gaussian noise and the normalized token codes, train a single time-conditioned MLP by flow matching on this path, and generate samples by integrating the learned ODE with an RK2 (Heun) solver. A linear coupling is the simplest admissible interpolant, but is sufficient since Proposition 1(ii) shows that decoder smoothness and non-degeneracy keeps the decoded trajectories nearly straight.

**Vectorization and normalization.** Let $\widehat{E} \in \mathbb{R}^{M \times d_e}$ be the decoder compatible token matrix produced by the autoencoder stage. We flatten tokens into a single vector

$$z_0 \;:=\; \text{vec}(\widehat{E}) \;\in\; \mathbb{R}^D, \qquad D \;=\; Md_e, \tag{11}$$

and apply a fixed affine normalization,

$$\tilde{z}_0 \;=\; \frac{z_0 - \bar{z}}{s},$$

where $\bar{z}$ is the dataset mean and $s > 0$ is a constant scale. The generator is trained *only* in this normalized space; sampling inverts the normalization before decoding.

**Linear coupling path and target velocity.** Having moved transport into this decoder compatible normalized token space, we instantiate flow matching with a linear coupling path. For each $\tilde{z}_0$, we draw $t \sim \text{Unif}[0,1]$ and $\varepsilon \sim \mathcal{N}(0, I_D)$, and define the straight path

$$x_t \;=\; (1-t)\varepsilon \;+\; t\tilde{z}_0, \tag{12}$$

whose *true* velocity is the constant vector

$$v^\star(x_t, t) \;=\; \frac{d}{dt}x_t \;=\; \tilde{z}_0 - \varepsilon. \tag{13}$$

TabFlowM parameterizes a time-dependent vector field $v_\theta : \mathbb{R}^D \times [0,1] \to \mathbb{R}^D$ with a single MLP that consumes $x_t$ and a Fourier (sin/cos) embedding of $t$.

**A more efficient training objective.** The benefit of equation 12–equation 13 is that it yields a *simulation-free, scale-stable* supervised target. Training tuples $(x_t, t, v^\star)$ are obtained in closed form from i.i.d. draws $(\tilde{z}_0, \varepsilon, t)$, so each gradient step requires only a single forward/backward pass through $v_\theta$. This contrasts with score-based diffusion objectives, which learn the time-indexed score $s^\star(x, t) = \nabla_x \log p_t(x)$ induced by a forward noising process. Under a common variance-exploding corruption $x_t = x_0 + \sigma(t)\varepsilon$, the conditional score obeys:

$$\nabla_{x_t} \log p(x_t \mid x_0) \;=\; -\frac{x_t - x_0}{\sigma(t)^2}, \tag{14}$$

so the effective regression target changes magnitude dramatically across $t$ (exploding as $\sigma(t) \to 0$ and vanishing for large $\sigma(t)$), motivating noise-dependent parameterizations, loss weights, and preconditioning to balance gradients. In contrast, the FM target $v^\star = \tilde{z}_0 - \varepsilon$ does not inherit any $\sigma(t)^{-2}$ amplification and remains a bounded displacement vector across time. Empirically, this produces a smoother optimization landscape and reduces tuning sensitivity, which is where we observe the largest efficiency gains.

**Flow matching loss.** We regress the velocity directly:

$$\mathcal{L}_{\text{FM}}(\theta) \;=\; \frac{1}{2} \, \mathbb{E}_{\tilde{z}_0, \varepsilon, t} \left\| v_\theta(x_t, t) \;-\; (\tilde{z}_0 - \varepsilon) \right\|_2^2. \tag{15}$$

Compared to diffusion-style training, equation 15 replaces score estimation, which is tightly coupled to the forward noising schedule and its time-dependent scaling, with direct velocity regression along an analytically specified coupling path. Geometrically, the loss encourages $v_\theta(x_t, t)$ to align with and match the magnitude of the displacement from $\varepsilon$ to $\tilde{z}_0$.

**Sampling (ODE integration).** At generation time we draw $x(0) \sim \mathcal{N}(0, I_D)$ and integrate the ODE

$$\frac{dx}{dt} \;=\; v_\theta\big(x(t), t\big), \quad t \in [0, 1], \quad x(0) \sim \mathcal{N}(0, I_D), \tag{16}$$

using Heun's method (RK2) with a fixed step grid $0 = t_0 < \cdots < t_S = 1$:

$$\begin{aligned}
x_{i+1}^{\text{Euler}} &= x_i + \Delta t \, v_\theta(x_i, t_i), \\
x_{i+1} &= x_i + \tfrac{\Delta t}{2} \big[ v_\theta(x_i, t_i) + v_\theta\big(x_{i+1}^{\text{Euler}}, t_{i+1}\big) \big].
\end{aligned} \tag{17}$$

**Reduced solver heuristics.** Both diffusion models and FM ultimately rely on evaluating a learned time-dependent field along a trajectory. However, diffusion sampling typically integrates reverse-time SDE/ODE dynamics whose stiffness and accuracy requirements can vary sharply across the noise schedule, often leading to many function evaluations (NFEs) and additional heuristics (e.g., step schedules and sometimes correction steps) to reach high fidelity. FM sampling instead integrates the learned velocity field directly, without recovering a score or multiplying by schedule-dependent factors (such as $\dot{\sigma}(t)\sigma(t)$ in probability-flow formulations), so the NFE–quality tradeoff is transparent: with a fixed-step RK2 solver, sampling cost is simply proportional to the chosen number of steps.

**Mixed-type decoding.** The de-normalized output $z^\star = s\, x(1) + \bar{z}$ is reshaped to $M \times d_e$ and passed through the per-column linear heads (Eq. 6): numerical values are recovered by affine projection and categorical values by arg max over softmax logits. The dataset's inverse transformers are then applied to obtain the final table.

Once synthesis has been moved into a decoder compatible normalized token space, the role of the generator is no longer to discover a complex transport geometry from scratch. Instead, it must accurately follow the target velocity along a simple coupling path toward the target token code. A crucial claim of TabFlowM is that, in this regularized setting, small pathwise velocity regression error is already sufficient to control the generated endpoint and drive the generated token-space distribution toward the target distribution in Wasserstein distance:

**Proposition 2 (Endpoint control from pathwise velocity regression)** *Under the linear-path FM surrogate, if the learned velocity field is Lipschitz in state, then small flow-matching regression error along the reference coupling path implies small endpoint error, and hence Wasserstein closeness between the generated token-space distribution and the target token-space distribution.*

---

**Algorithm 1** TabFlowM Training

---

**Require:** Dataset $\mathcal{T} = \{(x^{\text{num}}, x^{\text{cat}})\}$; VAE encoder $\text{Enc}_\phi$, transformer decoder $\text{Dec}_\psi$ with per-column linear heads $\{R_m\}$; velocity network $v_\theta$
 1: **Stage 1: Train the VAE** (Sec. 4.1)
 2: **for** each mini-batch $\{x_i\}$ **do**
 3:    Embed each column into a $d_e$-dimensional token and concatenate into token matrix $E_i$
 4:    Pass $E_i$ through the encoder to obtain posterior parameters $(\mu_i, \sigma_i)$
 5:    Sample $z_i = \mu_i + \sigma_i \odot \xi, \ \xi \sim \mathcal{N}(0, I)$
 6:    Decode $\hat{E}_i = \text{Dec}_\psi(z_i)$ and reconstruct each column via its linear head $R_m$
 7:    Update $\phi, \psi$ on $\ell_{\text{recon}} + \beta \, \text{KL}$ (Eq. 7) with adaptive $\beta$ (Eq. 10)
 8: **end for**
 9: **Stage 2: Train the flow-matching generator** (Sec. 4.2)
10: For every training row, extract the posterior mean $\mu_{i,1:}$ (no reparameterization) and pass it through the transformer decoder to obtain decoder-compatible tokens $\hat{E}_i = \text{Dec}_\psi(\mu_{i,1:})$ (Def. 1)
11: Flatten each $\hat{E}_i$ into a vector $z_{0,i} = \text{vec}(\hat{E}_i)$ and compute dataset mean $\bar{z}$ and scale $s$
12: **for** each mini-batch $\{z_{0,i}\}$ **do**
13:    Normalise: $\tilde{z}_{0,i} = (z_{0,i} - \bar{z})/s$
14:    Sample time $t \sim \text{Unif}[0, 1]$ and noise $\varepsilon \sim \mathcal{N}(0, I)$
15:    Form the interpolant $x_t = (1 - t)\,\varepsilon + t\,\tilde{z}_{0,i}$ (Eq. 12)
16:    Update $\theta$ to minimise $\|v_\theta(x_t, t) - (\tilde{z}_{0,i} - \varepsilon)\|^2$ (Eq. 15)
17: **end for**

---

**Algorithm 2** TabFlowM Sampling

---

**Require:** Trained velocity network $v_\theta$; per-column linear heads $\{R_m\}$ from the frozen decoder; dataset statistics $\bar{z}, s$; number of integration steps $S$
 1: Draw initial noise $x_0 \sim \mathcal{N}(0, I)$
 2: **for** $i = 0, \ldots, S - 1$ **do**
 3:    Advance $x_i \to x_{i+1}$ with one Heun (second-order Runge–Kutta) step using $v_\theta$
 4: **end for**
 5: De-normalise: $z^\star = s\,x_S + \bar{z}$
 6: Reshape $z^\star$ into an $M \times d_e$ token matrix $\hat{E}$
 7: Decode each column token $\hat{e}_m$ through its linear head $R_m$ to recover numerical values or categorical logits (Eq. 6)
 8: Apply inverse data transforms to obtain the synthetic row $\tilde{x}$
 9: **return** $\tilde{x}$

---

In other words, within the decoder compatible token space constructed above, optimizing equation 15 is sufficient to drive the generator toward the correct token-space law. Details of the proof including the regularity and numerical integration error are quantified in Appendix A.4.2. For clarity, we also provide pseudo code of TabFlowM's training and sampling procedure below in Algorithm 1 & 2.

## 5 Experiments

In this section, we present the methodologies and baselines used to benchmark TabFlowM against other existing general-purpose tabular-data synthesizers. Note that all baselines were tested five times with the average and standard deviation of results presented. All diffusion based methods have had their shared hyperparameter set equally to ensure fairness during evaluation. Specific hyperparameters for each model and hardware specifications are shown in Appendix A.5. Runtime comparisons use identical hardware and batch sizes across methods.

## 5.1 Baselines

We benchmark *TabFlowM* against classical methods such as CTGAN Xu et al. (2019), TVAE Xu et al. (2019), CopulaGAN Patki et al. (2016), and existing SOTA diffusion-based synthesizers TabSyn Zhang et al. (2024) and TabDiff Shi et al. (2024)[1]. Following the evaluation procedures in prior works Zhang et al. (2024); Shi et al. (2024), we assess synthethic quality through four axes: *Shape*, the average divergence between real and synthetic *marginal* column distributions; *Trend*, the discrepancy in pairwise dependence structure (via correlation or contingency-based statistics); *Machine Learning Efficiency* (column-wise MLE), which trains a predictor for each target column on synthetic data and tests on real data (AUC for classification; RMSE for regression) Zhang et al. (2024); Shi et al. (2024); and *C2ST-LR*, a logistic-regression two-sample test that quantifies distinguishability between real and synthetic rows. C2ST-LR is not a formal privacy guarantee but a distinguishability measure widely used as a heuristic proxy for memorization risk Zhang et al. (2024); Shi et al. (2024). The evaluation metrics used are introduced in Appendix A.2 with further evaluation settings specified in Appendix A.3 that varies from existing literature.

## 5.2 Datasets

We inherit the six real world tabular benchmarks used in prior work Shi et al. (2024); Zhang et al. (2024) from the UCI Machine Learning Repository Asuncion et al. (2007): *Adult*, *Default*, *Diabetes*, *Magic*, *News*, and *Beijing*. Across these datasets, the number of rows ranges from 768 (*Diabetes*) to 48,842 (*Adult*), and each benchmark contains a mix of numerical and categorical columns. *Adult*, *Default*, *Diabetes*, and *Magic* are treated as classification tasks, while *Beijing* and *News* are treated as regression tasks. In addition to these six controlled benchmarks, we also evaluate a large scale, rare event prediction stress test on the BAF fraud dataset, described separately in Section 6.2. For every dataset, we adopt a fixed 8:1:1 split into training, validation, and test sets. Details can be found in Appendix A.1.

# 6 Results

## 6.1 Controlled Benchmark Comparison

Table 1: Shape Score Results ($\downarrow$)

| Method | Adult | Beijing | Default | Diabetes | Magic | News |
|---|---|---|---|---|---|---|
| CTGAN | $0.192_{\pm0.016}$ | $0.230_{\pm0.012}$ | $0.107_{\pm0.007}$ | $0.125_{\pm0.008}$ | $0.077_{\pm0.029}$ | $0.186_{\pm0.007}$ |
| TVAE | $0.185_{\pm0.023}$ | $0.287_{\pm0.025}$ | $0.134_{\pm0.012}$ | $0.149_{\pm0.004}$ | $0.120_{\pm0.015}$ | $0.167_{\pm0.008}$ |
| CopulaGAN | $0.111_{\pm0.007}$ | $0.192_{\pm0.009}$ | $0.099_{\pm0.008}$ | $0.124_{\pm0.004}$ | $0.127_{\pm0.020}$ | $0.184_{\pm0.009}$ |
| TabSyn | $\underline{0.037}_{\pm0.004}$ | $0.067_{\pm0.011}$ | $0.079_{\pm0.030}$ | $\underline{0.048}_{\pm0.006}$ | $0.063_{\pm0.014}$ | $\mathbf{0.066}_{\pm0.007}$ |
| TabDiff | $0.043_{\pm0.015}$ | $\underline{0.044}_{\pm0.014}$ | $0.073_{\pm0.009}$ | $\mathbf{0.028}_{\pm0.005}$ | $0.046_{\pm0.008}$ | $0.076_{\pm0.016}$ |
| **TabFlowM** | $\mathbf{0.036}_{\pm0.008}$ | $\mathbf{0.040}_{\pm0.005}$ | $\mathbf{0.057}_{\pm0.014}$ | $0.057_{\pm0.010}$ | $\mathbf{0.036}_{\pm0.009}$ | $\underline{0.067}_{\pm0.008}$ |

Table 2: Trend Score Results ($\downarrow$)

| Method | Adult | Beijing | Default | Diabetes | Magic | News |
|---|---|---|---|---|---|---|
| CTGAN | $0.130_{\pm0.012}$ | $0.105_{\pm0.015}$ | $0.074_{\pm0.003}$ | $0.117_{\pm0.011}$ | $0.061_{\pm0.005}$ | $0.045_{\pm0.001}$ |
| TVAE | $0.093_{\pm0.009}$ | $0.198_{\pm0.030}$ | $0.112_{\pm0.018}$ | $0.203_{\pm0.002}$ | $0.045_{\pm0.010}$ | $0.028_{\pm0.002}$ |
| CopulaGAN | $0.112_{\pm0.010}$ | $0.112_{\pm0.009}$ | $0.078_{\pm0.003}$ | $0.157_{\pm0.011}$ | $0.054_{\pm0.003}$ | $0.045_{\pm0.001}$ |
| TabSyn | $\underline{0.051}_{\pm0.008}$ | $0.051_{\pm0.005}$ | $\underline{0.048}_{\pm0.013}$ | $0.087_{\pm0.010}$ | $\underline{0.019}_{\pm0.008}$ | $\mathbf{0.020}_{\pm0.002}$ |
| TabDiff | $0.065_{\pm0.018}$ | $\mathbf{0.038}_{\pm0.009}$ | $0.067_{\pm0.012}$ | $\mathbf{0.043}_{\pm0.006}$ | $0.068_{\pm0.036}$ | $0.032_{\pm0.006}$ |
| **TabFlowM** | $\mathbf{0.046}_{\pm0.016}$ | $\underline{0.043}_{\pm0.007}$ | $\mathbf{0.042}_{\pm0.014}$ | $0.104_{\pm0.015}$ | $\mathbf{0.013}_{\pm0.004}$ | $\underline{0.021}_{\pm0.001}$ |

We report **Shape** and **Trend** on the six benchmark datasets in Tables 1 and 2. **TabFlowM** is the strongest method on three of the six datasets, Adult, Default, and Magic, where it achieves the best result on both Shape and Trend but ranks first or second on both metrics for five of the six datasets. In aggregate, TabFlowM

---

[1]No working implementation of TabbyFlow Guzmán-Cordero et al. (2025) was available at the time of submission, and is therefore omitted. A conceptual comparison is provided in Appendix A.3.1.

achieves the lowest mean **Shape** ($\approx 0.049$) and the lowest mean **Trend** ($\approx 0.045$) across datasets. Across its wins, it improves over the next best method by an average of $\approx 15.6\%$, with gains as large as $\approx 31.6\%$ (Trend on *Magic*).

On **Shape**, TabFlowM is best on *Adult*, *Beijing*, *Default*, and *Magic*, with small but consistent gains on *Adult* ($\approx 2.7\%$) and larger gains on *Beijing* ($\approx 9.1\%$), *Default* ($\approx 21.9\%$), and *Magic* ($\approx 21.7\%$) relative to the next best method. On *News*, TabFlowM is essentially tied with the best method, trailing by only $\approx 1.5\%$. *Diabetes* remains the main outlier, where TabDiff is strongest.

On **Trend**, TabFlowM leads on *Adult*, *Default*, and *Magic*, improving over the next best method by $\approx 9.8\%$, $\approx 12.5\%$, and $\approx 31.6\%$, respectively. It is also competitive on *Beijing* and *News*, where it places second and is within $\approx 5.0\%$ of the best result on *News*.

Table 3: MLE Efficacy

| Method | Adult (AUC↑) | Beijing (RMSE↓) | Default (AUC↑) | Magic (AUC↑) | Diabetes (AUC↑) | News (RMSE↓) |
|---|---|---|---|---|---|---|
| CTGAN | $0.875_{\pm 0.007}$ | $1.053_{\pm 0.040}$ | $0.724_{\pm 0.015}$ | $0.872_{\pm 0.019}$ | $0.601_{\pm 0.036}$ | $0.945_{\pm 0.026}$ |
| TVAE | $0.878_{\pm 0.008}$ | $0.920_{\pm 0.038}$ | $0.716_{\pm 0.009}$ | $0.823_{\pm 0.021}$ | $0.614_{\pm 0.009}$ | $1.006_{\pm 0.040}$ |
| CopulaGAN | $0.859_{\pm 0.009}$ | $0.995_{\pm 0.031}$ | $0.647_{\pm 0.020}$ | $0.803_{\pm 0.009}$ | $0.570_{\pm 0.040}$ | $0.921_{\pm 0.009}$ |
| TabSyn | $\underline{0.899}_{\pm 0.003}$ | $\mathbf{0.709}_{\pm 0.017}$ | $0.702_{\pm 0.025}$ | $\underline{0.915}_{\pm 0.005}$ | $0.627_{\pm 0.021}$ | $0.863_{\pm 0.016}$ |
| TabDiff | $0.896_{\pm 0.007}$ | $\underline{0.716}_{\pm 0.049}$ | $\underline{0.741}_{\pm 0.025}$ | $0.890_{\pm 0.016}$ | $\underline{0.648}_{\pm 0.005}$ | $0.873_{\pm 0.030}$ |
| **TabFlowM** | $\mathbf{0.905}_{\pm 0.007}$ | $0.739_{\pm 0.012}$ | $\mathbf{0.747}_{\pm 0.020}$ | $\mathbf{0.918}_{\pm 0.003}$ | $\mathbf{0.651}_{\pm 0.021}$ | $\mathbf{0.837}_{\pm 0.016}$ |

Table 3 (**MLE Efficacy**) shows that TabFlowM achieves the best downstream performance on *Adult*, *Default*, *Magic*, *Diabetes*, and the *News* regression task, while remaining competitive on *Beijing*, where it is within $\sim 4.2\%$ of TabSyn in RMSE. On *News*, TabFlowM also obtains the lowest RMSE, improving over the next best method, TabSyn, by approximately 3.0%. Across all six datasets, TabFlowM attains the best average rank, indicating that its samples preserve task-relevant predictive structure at least as well as, and often better than, the strongest diffusion-based baselines.

Table 4: C2ST-LR

| Method | Adult | Beijing | Default | Diabetes | Magic | News |
|---|---|---|---|---|---|---|
| CTGAN | $0.712_{\pm 0.016}$ | $0.720_{\pm 0.015}$ | $0.742_{\pm 0.033}$ | $\underline{0.831}_{\pm 0.014}$ | $0.610_{\pm 0.052}$ | $0.653_{\pm 0.016}$ |
| TVAE | $0.803_{\pm 0.012}$ | $0.919_{\pm 0.048}$ | $0.823_{\pm 0.010}$ | $0.996_{\pm 0.001}$ | $0.713_{\pm 0.040}$ | $0.613_{\pm 0.029}$ |
| CopulaGAN | $0.678_{\pm 0.022}$ | $0.750_{\pm 0.013}$ | $0.766_{\pm 0.032}$ | $0.942_{\pm 0.013}$ | $0.722_{\pm 0.028}$ | $0.687_{\pm 0.025}$ |
| TabSyn | $\underline{0.607}_{\pm 0.017}$ | $0.620_{\pm 0.015}$ | $0.655_{\pm 0.021}$ | $0.887_{\pm 0.027}$ | $\mathbf{0.562}_{\pm 0.014}$ | $\mathbf{0.584}_{\pm 0.035}$ |
| TabDiff | $0.621_{\pm 0.037}$ | $\mathbf{0.580}_{\pm 0.046}$ | $\mathbf{0.605}_{\pm 0.038}$ | $\mathbf{0.598}_{\pm 0.020}$ | $0.662_{\pm 0.078}$ | $0.607_{\pm 0.120}$ |
| **TabFlowM** | $\mathbf{0.586}_{\pm 0.035}$ | $\underline{0.603}_{\pm 0.021}$ | $0.625_{\pm 0.052}$ | $0.903_{\pm 0.018}$ | $\underline{0.576}_{\pm 0.012}$ | $\underline{0.592}_{\pm 0.047}$ |

For **C2ST-LR** (Table 4), TabFlowM shows favorable distinguishability while delivering state-of-the-art fidelity. It achieves the best C2ST-LR on *Adult* and is a close second on four of the remaining five datasets: it is within $\approx 4.0\%$ of the best method on *Beijing* (0.603 vs. 0.580), $\approx 3.3\%$ on *Default* (0.625 vs. 0.605), $\approx 2.5\%$ on *Magic* (0.576 vs. 0.562), and $\approx 1.4\%$ on *News* (0.592 vs. 0.584). Averaged over datasets, TabFlowM is *second only to TabDiff* in mean C2ST-LR ($\approx 0.648$ vs. $\approx 0.612$), indicating that the substantial gains in sample quality are not accompanied by systematically worse distinguishability on these benchmarks.

Tables 5 and 6 report **training** and **sampling** times. TabFlowM is the fastest method for *training* on 5 of 6 datasets (*Adult*, *Default*, *Diabetes*, *Magic*, and *News*), and is competitive on *Beijing* (only $\approx 16.4\%$ slower than TabDiff while being $\approx 2.2\%$ faster than TabSyn). Importantly, it avoids the severe training-time blow-up exhibited by TabDiff on *Diabetes* (TabFlowM is $\approx 5.5\times$ faster).

Lastly, looking back at *Diabetes*, it should be noted that FM with a single linear coupling path is not uniformly best. Outside the idealized limit of infinite capacity and perfect optimization, TabDiff's richer backbone and adaptive schedules still win on Shape/Trend, suggesting that regimes with very sparse, highly skewed, or strongly multimodal categorical structure, more expressive transport paths could be beneficial. We explore this in section 7.

Table 5: Training Time (seconds; lower is better)

| Method | Adult | Beijing | Default | Diabetes | Magic | News |
|---|---|---|---|---|---|---|
| TabSyn | $\underline{61.08}_{\pm 15.10}$ | $70.64_{\pm 17.94}$ | $64.55_{\pm 15.25}$ | $\underline{128.84}_{\pm 39.47}$ | $42.21_{\pm 8.80}$ | $\underline{72.81}_{\pm 17.79}$ |
| TabDiff | $65.38_{\pm 4.64}$ | $\mathbf{59.37}_{\pm 2.71}$ | $\underline{60.69}_{\pm 3.05}$ | $686.93_{\pm 229.01}$ | $\underline{35.75}_{\pm 1.47}$ | $76.11_{\pm 6.29}$ |
| TabFlowM | $\mathbf{59.53}_{\pm 13.91}$ | $\underline{69.08}_{\pm 16.38}$ | $\mathbf{28.08}_{\pm 13.17}$ | $\mathbf{125.09}_{\pm 35.55}$ | $\mathbf{21.68}_{\pm 9.42}$ | $\mathbf{69.41}_{\pm 16.21}$ |

Table 6: Sampling Time (seconds; lower is better)

| Method | Adult | Beijing | Default | Diabetes | Magic | News |
|---|---|---|---|---|---|---|
| TabSyn | $\underline{6.32}_{\pm 1.21}$ | $\underline{6.063}_{\pm 1.07}$ | $\mathbf{3.04}_{\pm 0.05}$ | $\underline{10.49}_{\pm 1.38}$ | $\mathbf{1.76}_{\pm 0.04}$ | $6.18_{\pm 0.19}$ |
| TabDiff | $10.72_{\pm 1.73}$ | $8.49_{\pm 0.74}$ | $9.20_{\pm 0.95}$ | $488.44_{\pm 2.90}$ | $2.39_{\pm 0.69}$ | $11.31_{\pm 1.06}$ |
| TabFlowM | $\mathbf{4.07}_{\pm 0.95}$ | $\mathbf{4.35}_{\pm 1.27}$ | $\underline{3.47}_{\pm 1.10}$ | $\mathbf{9.52}_{\pm 0.21}$ | $\underline{1.96}_{\pm 1.12}$ | $\underline{6.44}_{\pm 1.01}$ |

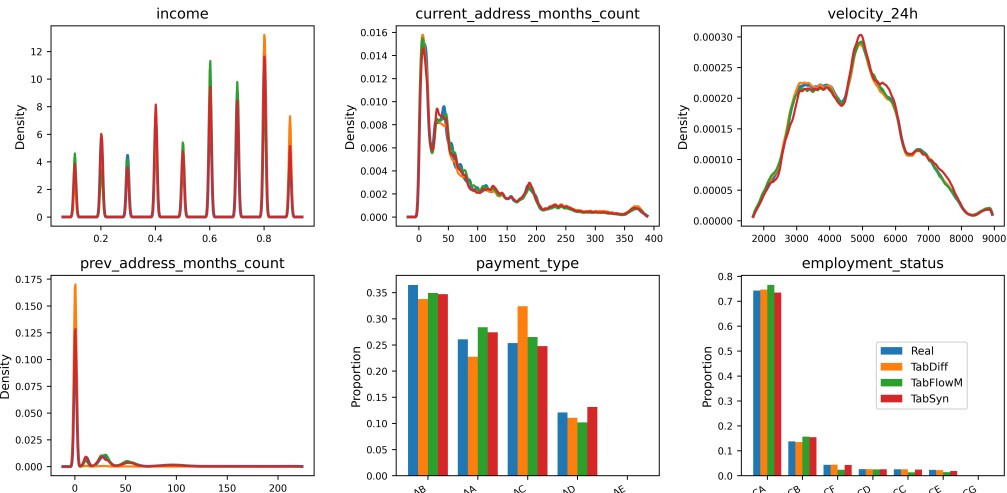

Figure 2: BAF Variant 1 Density Plots

## 6.2 Large-Scale Rare Event Prediction Stress Test: BAF (Kaggle)

Beyond the six controlled UCI benchmarks above, we evaluate a deployment-motivated stress test on the large scale, severely imbalanced BAF fraud dataset Jesus et al. (2022) (Appendix A.1). This setting is practically important because real world tabular synthesis often operates at substantially larger scale and must preserve structure relevant to rare event labels such as fraud. We therefore use BAF to evaluate whether methods that perform well on standard benchmarks remain effective under the combined challenges of scale and extreme class imbalance.

We evaluate six BAF configurations (BAF-Base and BAF-V1–V5), each representing a distinct preprocessing or variant setting within the same large scale, heavily imbalanced regime. Because ROC-AUC can be overly optimistic in rare event settings, we report **AUC-PR** as the downstream metric for BAF, since it focuses directly on precision-recall behavior for the minority fraud class. Given the computational footprint at this scale, we narrow the comparison to a focused study between diffusion-based methods (**TabDiff** and **TabSyn**) versus our flow matching model **TabFlowM**. Results are shown below:

On the large scale BAF fraud benchmarks, **TabFlowM** provides the strongest overall trade-off once unconditional fidelity, rare event predictive utility, and runtime are considered jointly. On **Shape**, the outcome is mixed: **TabFlowM** achieves the best result on **4 out of 6** variants (*V2, V3, V4,* and *V5*), while **TabSyn** is best on *BAF-Base* and *V1*. Averaged across all six variants, **TabFlowM** attains the lowest mean Shape of approximately 0.2349, followed closely by **TabSyn** at approximately 0.2379, with both methods clearly ahead of **TabDiff** at approximately 0.2509. On **Trend**, however, the picture is unambiguous: **TabFlowM** is best on **all six** variants and achieves the lowest mean Trend of approximately 0.1857, compared with 0.1952

Table 7: Large scale rare event prediction stress test on BAF (Kaggle)

| Dataset | Method | Shape↓ | Trend↓ | AUC-PR↑ | Train (s)↓ | Sample (s)↓ |
|---|---|---|---|---|---|---|
| BAF-Base | TabSyn | $\mathbf{0.2379}_{\pm 0.0035}$ | $\underline{0.2165}_{\pm 0.0105}$ | $\underline{0.1142}_{\pm 0.0163}$ | $\underline{5082.19}_{\pm 438.36}$ | $\mathbf{130.79}_{\pm 9.17}$ |
| | TabDiff | $0.2527_{\pm 0.0097}$ | $0.3806_{\pm 0.0153}$ | $0.0196_{\pm 0.0035}$ | $5179.24_{\pm 27.92}$ | $\underline{192.03}_{\pm 1.07}$ |
| | TabFlowM | $\underline{0.2381}_{\pm 0.0032}$ | $\mathbf{0.2124}_{\pm 0.0033}$ | $\mathbf{0.1226}_{\pm 0.0081}$ | $\mathbf{4795.51}_{\pm 21.02}$ | $195.62_{\pm 7.44}$ |
| BAF-V1 | TabSyn | $\mathbf{0.2366}_{\pm 0.0041}$ | $\underline{0.2149}_{\pm 0.0078}$ | $\underline{0.0936}_{\pm 0.0165}$ | $\underline{4427.60}_{\pm 471.61}$ | $\mathbf{132.22}_{\pm 18.03}$ |
| | TabDiff | $\underline{0.2479}_{\pm 0.0055}$ | $0.3480_{\pm 0.0533}$ | $0.0821_{\pm 0.0281}$ | $5231.47_{\pm 24.41}$ | $\underline{192.60}_{\pm 2.36}$ |
| | TabFlowM | $0.2612_{\pm 0.0031}$ | $\mathbf{0.2123}_{\pm 0.0031}$ | $\mathbf{0.1128}_{\pm 0.0186}$ | $\mathbf{3479.30}_{\pm 660.92}$ | $198.22_{\pm 4.95}$ |
| BAF-V2 | TabSyn | $\underline{0.2419}_{\pm 0.0044}$ | $\underline{0.1880}_{\pm 0.0080}$ | $\underline{0.1194}_{\pm 0.0204}$ | $\underline{4055.10}_{\pm 296.06}$ | $\mathbf{138.69}_{\pm 32.45}$ |
| | TabDiff | $0.2432_{\pm 0.0058}$ | $0.2917_{\pm 0.0640}$ | $0.0110_{\pm 0.0412}$ | $5234.40_{\pm 29.54}$ | $\underline{192.88}_{\pm 2.46}$ |
| | TabFlowM | $\mathbf{0.2313}_{\pm 0.0052}$ | $\mathbf{0.1725}_{\pm 0.0049}$ | $\mathbf{0.1233}_{\pm 0.0051}$ | $\mathbf{3503.80}_{\pm 537.48}$ | $195.57_{\pm 6.09}$ |
| BAF-V3 | TabSyn | $\underline{0.2400}_{\pm 0.0036}$ | $\underline{0.1952}_{\pm 0.0164}$ | $\underline{0.0854}_{\pm 0.0176}$ | $\underline{3918.31}_{\pm 264.18}$ | $\mathbf{136.90}_{\pm 25.81}$ |
| | TabDiff | $0.2463_{\pm 0.0055}$ | $0.3121_{\pm 0.0638}$ | $0.0775_{\pm 0.0320}$ | $5225.09_{\pm 27.32}$ | $192.90_{\pm 1.74}$ |
| | TabFlowM | $\mathbf{0.2263}_{\pm 0.0047}$ | $\mathbf{0.1740}_{\pm 0.0053}$ | $\mathbf{0.1279}_{\pm 0.0090}$ | $\mathbf{3506.36}_{\pm 537.52}$ | $\underline{189.93}_{\pm 14.80}$ |
| BAF-V4 | TabSyn | $\underline{0.2331}_{\pm 0.0029}$ | $\underline{0.1787}_{\pm 0.0081}$ | $\mathbf{0.1172}_{\pm 0.0250}$ | $\underline{5008.23}_{\pm 475.77}$ | $\mathbf{132.56}_{\pm 16.33}$ |
| | TabDiff | $0.2677_{\pm 0.0363}$ | $0.3592_{\pm 0.0429}$ | $0.0703_{\pm 0.0229}$ | $5207.32_{\pm 22.83}$ | $\underline{191.42}_{\pm 1.65}$ |
| | TabFlowM | $\mathbf{0.2263}_{\pm 0.0047}$ | $\mathbf{0.1713}_{\pm 0.0057}$ | $\underline{0.1157}_{\pm 0.0133}$ | $\mathbf{4582.58}_{\pm 526.32}$ | $194.42_{\pm 7.86}$ |
| BAF-V5 | TabSyn | $\underline{0.2380}_{\pm 0.0052}$ | $\underline{0.1779}_{\pm 0.0058}$ | $\underline{0.1085}_{\pm 0.0243}$ | $\underline{4667.96}_{\pm 516.26}$ | $\mathbf{126.84}_{\pm 8.51}$ |
| | TabDiff | $0.2476_{\pm 0.0090}$ | $0.2784_{\pm 0.0633}$ | $0.0750_{\pm 0.0267}$ | $5245.49_{\pm 32.65}$ | $\underline{192.36}_{\pm 2.12}$ |
| | TabFlowM | $\mathbf{0.2261}_{\pm 0.0047}$ | $\mathbf{0.1715}_{\pm 0.0056}$ | $\mathbf{0.1193}_{\pm 0.0152}$ | $\mathbf{4220.91}_{\pm 599.00}$ | $193.95_{\pm 9.05}$ |

for **TabSyn** and 0.3283 for **TabDiff**. This indicates that TabFlowM preserves cross column dependency structure more consistently than the diffusion baselines in this large scale, heavily imbalanced regime.

This advantage becomes even clearer on the task-aligned metric, **AUC-PR**. Because BAF is a rare event fraud benchmark, AUC-PR is the most relevant downstream criterion, as it directly measures predictive usefulness under severe class imbalance. Under this metric, **TabFlowM** ranks first on **5 out of 6** variants and is a very close second on *BAF-V4*, where **TabSyn** obtains 0.117 and **TabFlowM** obtains 0.116. Across all six variants, **TabFlowM** ranges from 0.113 to 0.128 and averages approximately 0.1195, compared with approximately 0.1063 for **TabSyn** and 0.0558 for **TabDiff**. A plausible explanation for the particularly weak minority class utility of **TabDiff** is its reliance on *learnable per-column noise schedules* Shi et al. (2024). In the severely imbalanced BAF setting, the gradients of these schedule parameters are dominated by majority class samples, so the learned schedules may become calibrated primarily to majority class conditional structure in each column. Minority class patterns, which occupy lower density and more weakly represented regions of the feature marginals, may therefore be under served during denoising. By contrast, **TabSyn** and **TabFlowM** do not learn separate per-column schedules, reducing the risk of such majority driven schedule specialization. This may help explain why both methods achieve stronger AUC-PR than **TabDiff**, even when unconditional fidelity alone does not fully reveal the gap.

Lastly, training remains costly for all approaches (thousands of seconds per run), reflecting the large scale setting. **TabFlowM** is the fastest to train in *every* variant, averaging 4015 s versus 5221 s for **TabDiff** and 4527 s for **TabSyn** (about 23% faster than **TabDiff** and 11% faster than **TabSyn** on average). Sampling exhibits the opposite pattern: **TabSyn** is the fastest sampler across variants (mean 133 s; range 126.84–138.69 s), while **TabDiff** and **TabFlowM** are similar (means 192 s and 195 s, respectively). Taken together, **TabFlowM** provides the strongest average *utility at scale*, with the best mean AUC-PR, competitive fidelity, the fastest training time, and sampling cost comparable to **TabDiff**.

# 7 Ablation Study

We now systematically isolate the main design choices in TabFlowM through a series of controlled experiments. All ablations share the same VAE weights, velocity MLP architecture, Heun (RK2) solver, and

training budget unless otherwise stated. In each experiment we vary exactly one factor: the coupling path geometry, the transport operating space, the solver budget, or the beta that.

## 7.1 Coupling Path Geometry and Transport Space

Table 8: Component ablations.

| | Adult | | | Magic | | |
|---|---|---|---|---|---|---|
| **Variant** | **Shape↓** | **Trend↓** | **MLE↑** | **Shape↓** | **Trend↓** | **MLE↑** |
| TabFlowM | $0.036_{\pm 0.008}$ | $\mathbf{0.046}_{\pm 0.016}$ | $\underline{0.905}_{\pm 0.007}$ | $0.036_{\pm 0.009}$ | $\mathbf{0.013}_{\pm 0.004}$ | $0.918_{\pm 0.021}$ |
| TabFlowM Quadratic | $0.030_{\pm 0.007}$ | $0.076_{\pm 0.009}$ | $0.898_{\pm 0.002}$ | $\underline{0.033}_{\pm 0.018}$ | $\underline{0.017}_{\pm 0.006}$ | $\underline{0.920}_{\pm 0.004}$ |
| TabFlowM Cosine | $\underline{0.030}_{\pm 0.004}$ | $\underline{0.060}_{\pm 0.007}$ | $0.900_{\pm 0.001}$ | $0.036_{\pm 0.014}$ | $0.023_{\pm 0.012}$ | $0.921_{\pm 0.004}$ |
| TabFlowM Z-space | $0.042_{\pm 0.008}$ | $0.088_{\pm 0.018}$ | $0.900_{\pm 0.004}$ | $0.042_{\pm 0.008}$ | $0.025_{\pm 0.018}$ | $0.912_{\pm 0.006}$ |
| TabFlowM no normalization | $\mathbf{0.014}_{\pm 0.009}$ | $0.079_{\pm 0.018}$ | $\mathbf{0.906}_{\pm 0.001}$ | $\mathbf{0.019}_{\pm 0.003}$ | $0.023_{\pm 0.004}$ | $\mathbf{0.929}_{\pm 0.005}$ |
| TabFlowM Token_dim 2 | $0.249_{\pm 0.053}$ | $0.414_{\pm 0.116}$ | $0.852_{\pm 0.027}$ | $0.242_{\pm 0.026}$ | $0.205_{\pm 0.069}$ | $0.604_{\pm 0.146}$ |
| TabFlowM Token_dim 8 | $0.056_{\pm 0.001}$ | $0.107_{\pm 0.003}$ | $0.896_{\pm 0.003}$ | $0.049_{\pm 0.004}$ | $0.027_{\pm 0.002}$ | $0.896_{\pm 0.003}$ |
| TabFlowM Token_dim 16 | $0.072_{\pm 0.008}$ | $0.159_{\pm 0.009}$ | $0.882_{\pm 0.011}$ | $0.069_{\pm 0.012}$ | $0.041_{\pm 0.006}$ | $0.909_{\pm 0.007}$ |

We start by comparing the default linear coupling path against a **quadratic** variant that replaces the interpolation coefficient $t$ with a slower-starting $t^2$ schedule, and a **cosine** variant that uses a smooth cosine schedule for the interpolation coefficient. These alternatives test whether TabFlowM's performance stems from the flow matching formulation in general, or whether the simple linear path is already well matched to the decoder compatible token space. We additionally evaluate **TabFlowM Z-space**, which shares the identical VAE weights, flow matching loss, velocity network, and hyperparameters with the default model. The only difference is the representation on which the flow model operates. TabFlowM trains on the decoder compatible token representation (Eq. 4), after which generation requires only the per-column linear heads (Eq. 6), whereas Z-space trains on the encoder-side latent (Eq. 3) and must pass generated samples through the full decoder transformer at sampling time. This comparison directly isolates the effect of the transport operating space from the generative backbone.

Finally, we ablate the **token dimension** $d_e$, i.e. the dimensionality of each per-column token embedding produced by the column tokenizer (Eq. 1). Because TabFlowM flattens the $M \times d_e$ token matrix before flow matching, the total transport dimensionality is $D = Md_e$. Therefore changing $d_e$ simultaneously affects the representational capacity of the autoencoder and the difficulty of the velocity regression problem faced by the downstream MLP.

Table 8 shows that the linear path remains the most balanced choice. The quadratic and cosine variants achieve broadly comparable Shape and MLE to the default, but both weaken Trend. This pattern is likely explained by the velocity target structure: the linear path produces a constant target $v^* = \tilde{z}_0 - \epsilon$, which a fixed-step RK2 solver integrates with minimal discretization error. Both quadratic and cosine targets vanish near $t = 0$ and ramp up later, producing a time-varying field that is harder to track with uniform step sizes. The resulting integration error likely affects directional accuracy along the trajectory more than endpoint magnitude, which would explain why pairwise structure (Trend) degrades while marginal fidelity (Shape) is largely preserved. The close similarity between quadratic and cosine is consistent with both sharing the same qualitative speed profile.

The Z-space variant on the other hand is consistently weaker in both fidelity metrics across both datasets. This is likely due to in decoder compatible space, the reconstruction map is per-column and linear, so Proposition 2's endpoint bound transfers cleanly to data space quality. In Z-space, generated tokens must pass through the full decoder transformer, whose cross-token attention couples all columns. Since the decoder was trained on real encoder outputs rather than FM generated tokens, small per-token errors can propagate across columns, degrading pairwise structure even when the flow itself is well trained. This supports the choice of decoder compatible token space as the primary transport interface.

The bottom three rows of Table 8 vary the per-column token dimension $d_e \in \{2, 8, 16\}$ while keeping all other architecture and training choices identical to the default ($d_e = 4$). Because the flow matching generator operates on the flattened representation of size $D = Md_e$, this sweep simultaneously controls two competing

effects: the per-column representational capacity of the autoencoder and the dimensionality of the transport problem that the velocity MLP must solve.

The sensitivity to $d_e$ is pronounced. At the low end, $d_e = 2$ collapses almost entirely: Shape exceeds 0.24 and Trend exceeds 0.20 on both *Adult* and *Magic*, while MLE drops sharply (to 0.852 on *Adult* and 0.604 on *Magic*). Two dimensions per token are evidently insufficient to encode the per-column information required for faithful reconstruction; the autoencoder bottleneck dominates all downstream metrics regardless of how well the flow model is trained. At the default $d_e = 4$, joint fidelity is strongest: Shape and Trend reach their best combined values, and MLE peaks or nearly peaks on both datasets. Increasing the token dimension to $d_e = 8$ or $d_e = 16$, however, progressively degrades all three metrics. On *Adult*, Shape rises from 0.036 to 0.056 ($d_e$=8) and 0.072 ($d_e$=16); Trend deteriorates from 0.046 to 0.107 and 0.159, respectively. The same monotonic degradation appears on *Magic*. This pattern suggests that higher-dimensional token spaces inflate the transport dimensionality $D$ without a commensurate gain in representational power, making the velocity field harder to learn under the fixed MLP capacity and training budget used across all experiments. In effect, the additional latent dimensions introduce redundancy that the flow model must learn to ignore, widening the gap between the optimization landscape and the information content of the representation.

Taken together, the token dimension sweep highlights a capacity to transport trade-off that is specific to latent flow matching architectures: $d_e$ must be large enough for the autoencoder to produce a decodable continuous representation, yet small enough that the resulting transport space remains tractable for a lightweight velocity network. The default $d_e = 4$ sits at the favorable balance point for the datasets and model sizes considered here.

## 7.2 Sampling Efficiency under Controlled Solver Budgets

To verify that TabFlowM's runtime advantages (Tables 5 and 6) reflect genuine algorithmic efficiency rather than merely using fewer solver evaluations at test time, we conduct a controlled sweep over the Number of Function Evaluations (NFE) in $\{4, 10, 20, 40, 100\}$. All methods use the same RK2 (Heun) solver and step schedules with only the number of integration steps being different. We report **Quality** as the average of the Shape and Trend error rates. Since both quantities are errors, lower Quality indicates better joint fidelity across marginal distributions and pairwise dependencies. We include TabFlowM Z-space in this sweep so that the solver budget comparison also tests whether the proposed decoder compatible token space remains preferable under matched integration budgets.

Table 9: NFE sweep.

| Method | News Quality↓ | | | | | Adult Quality↓ | | | | | BAF-Base Quality↓ | | | | |
|---|---|---|---|---|---|---|---|---|---|---|---|---|---|---|---|
| | 4 | 10 | 20 | 40 | 100 | 4 | 10 | 20 | 40 | 100 | 4 | 10 | 20 | 40 | 100 |
| TabFlowM | 0.124 | 0.060 | **0.048** | **0.045** | 0.044 | 0.365 | 0.091 | **0.052** | **0.047** | **0.041** | 0.308 | **0.242** | **0.231** | **0.230** | **0.225** |
| TabSyn | 0.287 | 0.179 | 0.057 | 0.053 | **0.043** | 0.612 | 0.294 | 0.070 | 0.053 | 0.044 | 0.720 | 0.343 | 0.248 | 0.235 | 0.227 |
| TabDiff | 0.277 | 0.143 | 0.076 | 0.063 | 0.054 | 0.531 | 0.123 | 0.071 | 0.058 | 0.054 | 0.605 | 0.443 | 0.365 | 0.353 | 0.317 |
| TabFlowM Z-space | **0.082** | **0.057** | 0.060 | 0.058 | 0.058 | **0.106** | **0.069** | 0.067 | 0.067 | 0.066 | **0.284** | 0.245 | 0.242 | 0.242 | 0.240 |

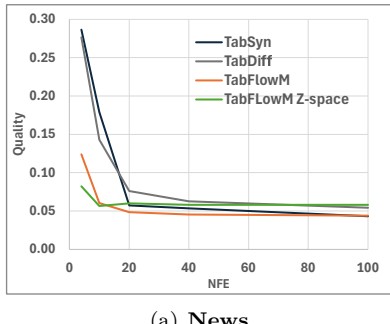
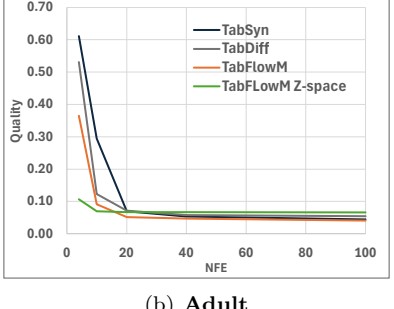
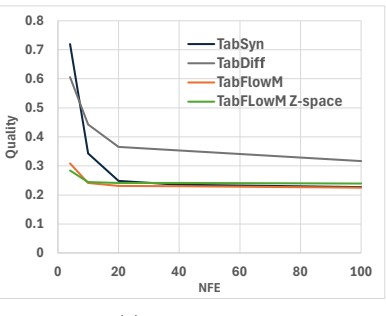

(a) **News**        (b) **Adult**        (c) **BAF-Base**

Figure 3: Sampling efficiency under matched solver budges on News, Adult, and BAF Base

Table 9 and Figure 3 show that the flow matching variants converge to near-peak Quality with substantially fewer function evaluations than the diffusion baselines. On Adult, TabFlowM achieves the strongest Quality

at every budget and reaches 0.041 at NFE= 100, compared with 0.044 for TabSyn and 0.054 for TabDiff. On BAF-Base, it is again consistently strongest, reaching 0.225 compared with 0.227 for TabSyn and 0.317 for TabDiff. On News, it substantially improves over TabDiff throughout the sweep while closely matching TabSyn at high NFE. The separation is most visible at low and moderate budgets: at NFE= 10 on Adult, TabFlowM already reaches 0.091, well below TabSyn (0.294) and TabDiff (0.531). This confirms that the efficiency gain is intrinsic to the flow matching formulation. Specifically, the bounded, time-homogeneous velocity target (Eq. 15) produces a smoother ODE field that an RK2 solver can track accurately with fewer steps.

Interestingly, the comparison between TabFlowM and Z-space across the NFE sweep further reveals a characteristic crossover that clarifies the role of the transport interface. At very low NFE ($\leq 4$), the Heun integrator is coarse, and Z-space benefits because its generated samples still pass through the full decoder transformer at no additional NFE cost. The transformer's cross-token attention can act as an implicit nonlinear corrector that can partially smooth integration errors. As NFE grows, however, this advantage reverses. In the decoder compatible token space, a more accurate flow output passes through only the per-column linear heads, so quality improves monotonically with integration precision. In Z-space, the generated encoder tokens must still traverse the decoder transformer, whose attention layers were trained on real encoder outputs and may introduce correlated cross-column error when applied to generated tokens that are slightly off the training distribution. This pattern is consistent across all three evaluation settings. Z-space saturates at a quality floor that is strictly higher than TabFlowM's converged value: $\sim 0.058$ versus $\sim 0.044$ on News, $\sim 0.066$ versus $\sim 0.041$ on Adult, and $\sim 0.240$ versus $\sim 0.225$ on BAF-Base. In each case, additional NFE beyond the crossover point (approximately NFE= 10 depending on the dataset) improves TabFlowM but yields no further gain for Z-space, suggesting that the quality ceiling is set by the decoder transformer's off-distribution sensitivity rather than by insufficient integration precision. This supports the choice of decoder compatible token transport for the main model.

### 7.3 Effect of the Adaptive $\beta$ Schedule

A key VAE training detail in TabFlowM is the adaptive $\beta$ schedule (Eq. 9–10), which initially applies moderate KL pressure to regularize the latent geometry and then relaxes it once reconstruction loss has plateaued, allowing the decoder-compatible token representation to retain finer record-level structure. To test whether this schedule is necessary, we compare against several fixed annealing settings. We conduct this ablation on the BAF-Base fraud benchmark rather than the UCI datasets for two reasons. First, the adaptive decay (Eq. 9–10) is triggered only when the per-column reconstruction loss stagnates beyond a patience window; on the smaller UCI benchmarks, training converges quickly enough that the schedule rarely fires, making the two $\beta$ regimes nearly indistinguishable in practice. BAF's million-row scale and severe class imbalance ($\sim 100{:}1$) produce a longer, more heterogeneous optimization trajectory in which the stagnation criterion is reached repeatedly, giving the adaptive schedule room to materially diverge from the fixed baseline. Second, the higher training budget required at this scale amplifies any downstream effect of the latent geometry on rare event structure, making BAF the most sensitive test bed for this ablation. Both variants use the same VAE architecture, training budget, and downstream flow matching configuration; only the $\beta$ schedule differs.

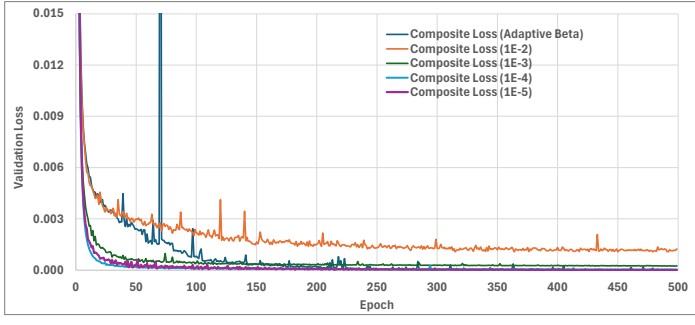

Table 10: Ablation of the VAE annealing on BAF-Base.

| Annealing | Shape↓ | Trend↓ | AUC-PR↑ |
|---|---|---|---|
| Adaptive | 0.2306 | **0.2090** | 0.1226 |
| $10^2$ | 0.2457 | 0.2255 | 0.0777 |
| $10^3$ | **0.2303** | 0.3625 | 0.1284 |
| $10^4$ | 0.2307 | 0.2103 | 0.0790 |
| $10^5$ | 0.2308 | 0.2105 | **0.1330** |

Figure 4: Validation loss under different $\beta$ values

Figure 4 shows the composite validation loss over training epochs. The adaptive schedule provides a stable optimization trajectory by initially regularizing the latent space and then relaxing the KL pressure once reconstruction improvement has plateaued. Table 10 quantifies the downstream impact across several annealing settings on BAF-Base. The adaptive schedule achieves the best Trend score (0.2090) and remains essentially tied for the best Shape score (0.2306 versus 0.2303 for the best fixed setting). While the $10^5$ setting obtains the highest AUC-PR (0.1330), the adaptive schedule remains competitive (0.1226) while preserving stronger aggregate dependency fidelity. In contrast, some fixed settings improve one metric only by sacrificing another: for example, $10^3$ gives the best Shape score but substantially worsens Trend (0.3625), indicating degraded pairwise structure despite strong marginal fidelity. Similarly, $10^2$ performs worse across all three metrics relative to the adaptive schedule. These results suggest that the adaptive $\beta$ schedule offers the most balanced latent geometry, preserving marginal fidelity, pairwise structure, and rare-event utility without manually tuning a fixed annealing scale. Since the ideal $\beta$ can vary with dataset scale, sparsity, imbalance, and reconstruction difficulty, we set $\beta$ adaptively to remove one less dataset specific hyperparameter.

### 7.4 Flow Matching Applied to other Diffusion Pipelines

Table 11: Consolidated results for **cflow**, **TabDiff**, and **TabFlowM**

| Dataset | Method | Shape↓ | Trend↓ | MLE | Train (s)↓ | Sample (s)↓ |
|---|---|---|---|---|---|---|
| Adult | cflow | $\mathbf{0.035}_{\pm0.007}$ | $\underline{0.059}_{\pm0.006}$ | $0.887_{\pm0.002}$ | $\underline{64.99}_{\pm4.42}$ | $\underline{7.83}_{\pm1.25}$ |
| | TabDiff | $0.043_{\pm0.015}$ | $0.065_{\pm0.018}$ | $\underline{0.896}_{\pm0.007}$ | $65.38_{\pm4.64}$ | $10.72_{\pm1.73}$ |
| | TabFlowM | $\underline{0.036}_{\pm0.008}$ | $\mathbf{0.046}_{\pm0.016}$ | $\mathbf{0.905}_{\pm0.007}$ | $\mathbf{59.53}_{\pm13.91}$ | $\mathbf{4.07}_{\pm0.95}$ |
| Default | cflow | $0.084_{\pm0.014}$ | $0.155_{\pm0.010}$ | $0.713_{\pm0.014}$ | $\underline{59.10}_{\pm2.75}$ | $\underline{3.73}_{\pm0.10}$ |
| | TabDiff | $\underline{0.073}_{\pm0.009}$ | $\underline{0.067}_{\pm0.012}$ | $\underline{0.741}_{\pm0.025}$ | $60.69_{\pm3.05}$ | $9.20_{\pm0.95}$ |
| | TabFlowM | $\mathbf{0.057}_{\pm0.014}$ | $\mathbf{0.042}_{\pm0.014}$ | $\mathbf{0.747}_{\pm0.020}$ | $\mathbf{28.08}_{\pm13.17}$ | $\mathbf{3.47}_{\pm1.10}$ |
| Diabetes | cflow | $\underline{0.040}_{\pm0.006}$ | $\underline{0.058}_{\pm0.008}$ | $0.613_{\pm0.010}$ | $\underline{452.74}_{\pm148.50}$ | $\underline{250.80}_{\pm1.64}$ |
| | TabDiff | $\mathbf{0.028}_{\pm0.005}$ | $\mathbf{0.043}_{\pm0.006}$ | $\underline{0.649}_{\pm0.005}$ | $686.93_{\pm229.01}$ | $488.44_{\pm2.90}$ |
| | TabFlowM | $0.057_{\pm0.010}$ | $0.104_{\pm0.015}$ | $\mathbf{0.651}_{\pm0.021}$ | $\mathbf{125.09}_{\pm35.55}$ | $\mathbf{9.52}_{\pm0.21}$ |
| Magic | cflow | $\mathbf{0.034}_{\pm0.008}$ | $0.112_{\pm0.006}$ | $0.831_{\pm0.028}$ | $\underline{35.10}_{\pm1.87}$ | $\mathbf{1.25}_{\pm0.08}$ |
| | TabDiff | $0.046_{\pm0.008}$ | $\underline{0.068}_{\pm0.036}$ | $\underline{0.890}_{\pm0.016}$ | $35.75_{\pm1.47}$ | $2.39_{\pm0.69}$ |
| | TabFlowM | $\underline{0.036}_{\pm0.009}$ | $\mathbf{0.013}_{\pm0.004}$ | $\mathbf{0.919}_{\pm0.003}$ | $\mathbf{21.68}_{\pm9.42}$ | $\underline{1.96}_{\pm1.12}$ |
| Beijing | cflow | $\mathbf{0.034}_{\pm0.006}$ | $0.079_{\pm0.013}$ | $0.811_{\pm0.063}$ | $\underline{60.80}_{\pm3.62}$ | $\mathbf{3.60}_{\pm0.69}$ |
| | TabDiff | $0.044_{\pm0.014}$ | $\mathbf{0.038}_{\pm0.009}$ | $\mathbf{0.716}_{\pm0.049}$ | $\mathbf{59.37}_{\pm2.71}$ | $8.49_{\pm0.74}$ |
| | TabFlowM | $\underline{0.040}_{\pm0.005}$ | $\underline{0.043}_{\pm0.007}$ | $\underline{0.739}_{\pm0.012}$ | $69.08_{\pm16.38}$ | $\underline{4.35}_{\pm1.27}$ |
| News | cflow | $\mathbf{0.052}_{\pm0.006}$ | $0.038_{\pm0.001}$ | $0.949_{\pm0.023}$ | $\underline{71.73}_{\pm5.70}$ | $\underline{7.55}_{\pm1.23}$ |
| | TabDiff | $0.076_{\pm0.016}$ | $\underline{0.032}_{\pm0.006}$ | $\underline{0.873}_{\pm0.030}$ | $76.11_{\pm6.29}$ | $11.31_{\pm1.06}$ |
| | TabFlowM | $\underline{0.067}_{\pm0.008}$ | $\mathbf{0.021}_{\pm0.001}$ | $\mathbf{0.837}_{\pm0.016}$ | $\mathbf{69.41}_{\pm16.21}$ | $\mathbf{6.44}_{\pm1.01}$ |

To test whether flow matching still helps when combined with a more sophisticated numerical backbone, we introduce **cflow**, which reuses TabDiff's continuous-time transformer and per-feature (column) adaptive noise schedules for numerical features, but is entirely retrained by replacing the EDM score objective with a FM loss. Table $8^2$ shows that TabDiff's richer architecture, the joint multi-modal diffusion with learned per-feature schedules, indeed increases modelling power: on harder regimes such as *Diabetes* it delivers the strongest Shape/Trend scores, while cflow remains competitive on several fidelity and runtime metrics. However, this expressivity comes at a substantial computational cost: both TabDiff and cflow remain much slower to sample than TabFlowM on several datasets, and on *Diabetes* they are also far more expensive to train. By contrast, the minimalist TabFlowM latent generator attains the best overall balance: it matches or surpasses TabDiff and cflow on four of six datasets in Shape/Trend (*Adult*, *Default*, *Magic*, *News*), achieves the best MLE on five of six datasets, including the lowest *News* RMSE of $0.837_{\pm0.016}$, and does so with

---

[2]Shape/Trend and times: lower is better. MLE is AUC↑ except *Beijing* (RMSE↓) and *News* (RMSE↓).

consistently lower training and sampling times. This ablation therefore suggests that TabDiff-style per-feature schedulers can be combined with flow matching to produce a competitive heavier backbone, but that most of the practical benefits can be captured by a lean FM design in the TabSyn-style latent space.

## 8 Conclusion

We have introduced TabFlowM, a minimalist flow matching framework operating in the latent space of a transformer VAE for mixed-type tabular data. Experiments on six public benchmarks show that TabFlowM matches or surpasses strong diffusion based baselines in Shape, Trend, and column-wise MLE metrics, while achieving competitive C2ST-LR and markedly lower training and sampling cost. This performance extends to the BAF fraud dataset with class imbalance, where TabFlowM achieves the best MLE/fraud detection performance among the compared diffusion-based methods. Taken together, these results suggest that once a decoder-compatible latent transport space has been constructed, deterministic flow matching captures most of the practical benefits of diffusion without requiring its full score-based framework. Ablation studies further indicate that a single FM generator in a shared latent space captures most of the practical benefits of heavier diffusion backbones, highlighting transport design rather than architectural complexity as a key driver of performance.

A limitation of TabFlowM is that its minimalist flow matching design relies on a single global transport path in the learned latent space, which may underfit highly multimodal regions or regimes with extreme categorical sparsity. Moreover, model performance depends on the VAE's ability to construct a well behaved transport space. Future work will therefore investigate feature conditioned velocity fields, richer path geometries, and more adaptive latent transport mechanisms to better handle such challenging regimes.

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

## A Appendix

### A.1 Dataset Details

This section specifies the details of the datasets used for evaluation.

**UCI Machine Learning Repository**

| Dataset | # Rows | # Num | # Cat | # Train | # Validation | # Test | Task |
|---------|--------|-------|-------|---------|--------------|--------|------|
| **Adult** | 48,842 | 6 | 9 | 28,943 | 3,618 | 16,281 | Classification |
| **Default** | 30,000 | 14 | 11 | 24,000 | 3,000 | 3,000 | Classification |
| **Diabetes** | 768 | 7 | 2 | 614 | 77 | 77 | Classification |
| **Magic** | 19,019 | 10 | 1 | 15,215 | 1,902 | 1,902 | Classification |
| **Beijing** | 43,824 | 7 | 5 | 35,058 | 4,383 | 4,383 | Regression |
| **News** | 39,644 | 46 | 2 | 31,714 | 3,965 | 3,965 | Regression |

**Bank Account Fraud Dataset Suite (BAF)**

The *Bank Account Fraud (BAF) Dataset Suite* Jesus et al. (2022), published at NeurIPS 2022, is a widely used open-source collection of six synthetic tabular datasets released on Kaggle. It is specifically designed to benchmark machine learning models for *bank account opening fraud detection* and is often regarded as a realistic, privacy-preserving, and challenging testbed for detecting *synthetic identity fraud*. All variants of the BAF dataset contain 1,000,000 entries and 32 columns, comprising 17 categorical and 15 numerical features. The imbalance ratio across each variant is also on average 100:1.

## A.2  Evaluation metrics

Let $\mathcal{N}$ and $\mathcal{C}$ denote the index sets of numerical and categorical columns, respectively, and let $d = |\mathcal{N}| + |\mathcal{C}|$.

**Shape.** For every column $j$ we compare the empirical marginal distributions of real and synthetic data with

$$\text{Shape} = \frac{1}{d} \sum_{j=1}^{d} \begin{cases} \text{KST}\big(X_j^{\text{real}}, X_j^{\text{syn}}\big), & j \in \mathcal{N}, \\ \text{TVD}\big(X_j^{\text{real}}, X_j^{\text{syn}}\big), & j \in \mathcal{C}, \end{cases}$$

where KST denotes the two-sample Kolmogorov–Smirnov statistic and Total-variation distance TVD between empirical probability vectors. Shape $\in [0, 1]$ with lower values indicating closer agreement.

**Trend.** For every same-type column pair $(j, k)$ we measure the discrepancy of second-order statistics:

$$\text{Trend} = \frac{1}{|\{(j,k)\}|} \sum_{\substack{j < k \\ (j,k) \text{ same type}}} \begin{cases} \frac{1}{2}\big|\rho_{jk}^{\text{real}} - \rho_{jk}^{\text{syn}}\big|, & j, k \in \mathcal{N}, \\ \frac{1}{2} \sum_{a,b}\big|P_{jk}^{\text{real}}(a, b) - P_{jk}^{\text{syn}}(a, b)\big|, & j, k \in \mathcal{C}. \end{cases}$$

Pairs mixing numerical and categorical attributes are ignored, following TabDiff's original definition.

**Column-wise MLE.** We evaluate conditional fidelity by training a separate downstream predictor for each target column. For every feature $x_j$, we treat the remaining attributes $x_{\setminus j}$ as inputs and train an XGBoost model on the synthetic dataset to predict $x_j$. The trained model is then evaluated on a held-out 20% split of real data, quantifying how well relationships learned from synthetic samples generalize to the true distribution. For categorical targets, we employ binary or multiclass objectives and report the area under the receiver operating characteristic curve (AUC ↑), while for numerical targets we use a regression objective and report the root-mean-square error (RMSE ↓). Averaging these per-column scores yields dataset-level indicators of conditional dependency fidelity between real and synthetic data.

**C2ST-LR.** Classifier Two-Sample Test with Logistic Regression: we label real rows as 1 and synthetic rows as 0, train a logistic regression discriminator, and report the ROC-AUC on a held-out split. Values close to 0.5 indicate that real and synthetic samples are hard to distinguish (better), while higher values indicate greater distinguishability (worse). We average the ROC-AUC over five random train/test splits.

### A.3 Further Evaluation Details

For all baselines, we use a common training budget of 100 epochs for UCI datasets and 500 epochs for the BAF datasets. Our goal is not to reproduce each method's original long run configuration exactly, but to compare methods under a unified and practical optimization budget. This controlled budget keeps runtime comparisons interpretable across heterogeneous model families, avoids large disparities in training duration across methods, and makes large scale experiments such as BAF tractable. To show that each model is trained sufficiently we show the training validation loss for Tabdiff and TabSyn's VAE in Fig.3 for News, the largest dataset among the UCI machine learning repository datasets.

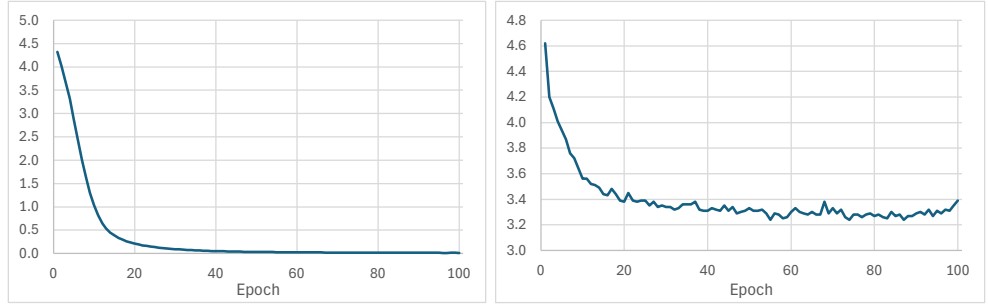

Figure 5: News Dataset Validation Loss for Tabsyn (VAE) (left) and TabDiff (right)

Interestingly, under our unified preprocessing/splitting/evaluation pipeline, the 100 epoch versions of TabSyn and TabDiff achieve **improved** fidelity scores while exhibiting **lower** variance. This suggests that much of their practical performance may already emerge under a moderate training budget.

#### A.3.1 Conceptual Comparison with TabbyFlow

TabbyFlow's exponential-family perspective on flow matching was an inspiration for this work: it motivated us to investigate whether a simpler, unified flow matching formulation in latent space could achieve competitive fidelity without per-column distributional choices. However, ambiguity in areas that materially affect performance existed such as around the functional form of the time dependent covariance in the Gaussian likelihood for numerical features, the parameterization of the interpolation path, and the noise model used for categorical interpolation in one-hot space. We therefore restrict our discussion to a conceptual comparison and leave a controlled empirical study to future work.

**Conceptual comparison of TabFlowM and TabbyFlow.**

| Design Axis | TabFlowM | TabbyFlow |
|---|---|---|
| Transport space | Learned latent token space (VAE) | Raw data space |
| Categorical handling | Unified: categoricals are embedded into continuous tokens; FM operates on a homogeneous space | Per-column: each categorical column has its own exponential-family likelihood (cross-entropy) |
| Numerical loss | Standard velocity MSE | Gaussian log-likelihood with time-dependent covariance |
| Velocity target | Constant: $v^\star = \tilde{z}_0 - \varepsilon$ | Implicit: derived from predicted sufficient statistics |
| Sampling | Heun (RK2) on a fixed grid | Adaptive ODE solver |

### A.4 Proofs

#### A.4.1 Geometric Justification for Transport in decoder compatible Token Space (Proposition 1)

This section formalizes the geometric motivation for performing flow matching in the learned decoder compatible token space rather than directly in raw mixed-type coordinates. The first part of the result shows

that straight interpolation in one-hot categorical space immediately leaves the set of valid categorical states. The second part shows that, under mild local regularity of the reconstruction map, straight segments in the continuous token representation decode to trajectories with controlled bending. Together, these two observations justify why simple transport paths are substantially more natural in this learned representation than in the original mixed-type space.

**Proposition 1 (Raw-space mismatch versus decoder compatible token-space regularity)**
*Straight interpolation is geometrically mismatched in raw one-hot space, whereas in decoder compatible token space it decodes to trajectories with uniformly controlled bending under mild local regularity.*

More precisely, consider two records that differ only in one categorical column $j$, whose one-hot encodings are $e_a, e_b \in \{0,1\}^{C_j}$ with $a \neq b$, where $C_j$ is the number of categories in column $j$.

**(Raw-space mismatch).** For any $t \in (0,1)$,

$$(1-t)e_a + te_b \in \Delta^{C_j-1} \setminus \{e_1, \ldots, e_{C_j}\},$$

so the straight segment between $e_a$ and $e_b$ leaves the set of valid categorical states. Hence straight Euclidean interpolation in raw one-hot space does not remain on the discrete data manifold.

**(decoder compatible token-space regularity).** Let $R : \mathbb{R}^D \to \mathbb{R}^q$ denote the reconstruction map from flattened decoder compatible token representations to decoded outputs, where $D = Md_e$ is the dimension of the flattened token representation and $q$ is the ambient dimension of the decoded output space. Let

$$\gamma(t) = (1-t)r_1 + tr_2, \qquad t \in [0,1],$$

be a straight segment between two token-space points $r_1, r_2 \in \mathbb{R}^D$, and define the decoded trajectory

$$x(t) = R(\gamma(t)).$$

Assume that $R$ is twice continuously differentiable on a convex region containing $\gamma([0,1])$, and that for all $u$ on this segment,

$$\|H_R(u)\|_{\text{op}} \leq \beta, \qquad \sigma_{\min}(J_R(u)) \geq m > 0,$$

where $J_R(u)$ and $H_R(u)$ denote the Jacobian and Hessian of $R$ at $u$, respectively. Then

$$\|\dot{x}(t)\|_2 \geq m \|r_2 - r_1\|_2, \qquad \|\ddot{x}(t)\|_2 \leq \beta \|r_2 - r_1\|_2^2,$$

and therefore

$$\frac{\|\ddot{x}(t)\|_2}{\|\dot{x}(t)\|_2^2} \leq \frac{\beta}{m^2}.$$

Thus the decoded image of a straight token-space segment has uniformly controlled normalized bending.

*Proof.* We prove the two parts in turn.

For the raw-space statement, let $e_a$ and $e_b$ be distinct one-hot vectors. For any $t \in (0,1)$, the vector

$$(1-t)e_a + te_b$$

has two strictly positive entries, namely $1-t$ in coordinate $a$ and $t$ in coordinate $b$, and zeros elsewhere. Hence it lies in the simplex $\Delta^{C_j-1}$ but is not itself a vertex, so it is not a valid categorical state.

For the token-space statement, differentiate the decoded path $x(t) = R(\gamma(t))$ along the straight segment $\gamma(t) = (1-t)r_1 + tr_2$. By the chain rule,

$$\dot{x}(t) = J_R(\gamma(t))(r_2 - r_1), \tag{18}$$

and

$$\ddot{x}(t) = H_R(\gamma(t))[r_2 - r_1, r_2 - r_1]. \tag{19}$$

The lower bound on the smallest singular value of $J_R(\gamma(t))$ yields

$$\|\dot{x}(t)\|_2 = \|J_R(\gamma(t))(r_2 - r_1)\|_2 \geq m \|r_2 - r_1\|_2.$$

Similarly, the Hessian bound implies

$$\|\ddot{x}(t)\|_2 = \|H_R(\gamma(t))[r_2 - r_1, r_2 - r_1]\|_2 \leq \beta \|r_2 - r_1\|_2^2.$$

Combining the two inequalities gives

$$\frac{\|\ddot{x}(t)\|_2}{\|\dot{x}(t)\|_2^2} \leq \frac{\beta \|r_2 - r_1\|_2^2}{m^2 \|r_2 - r_1\|_2^2} = \frac{\beta}{m^2}.$$

Proposition 1 isolates the geometric contrast used in the main text: raw mixed-type coordinates are poorly suited to straight interpolation, whereas decoder compatible token coordinates admit simple paths whose decoded trajectories remain well behaved under local regularity. This supports moving transport into the learned continuous representation, after which a lightweight path choice can be introduced in the main method.

**Realism of the regularity assumptions.** The assumptions of Proposition 1, a uniformly bounded Hessian $\|H_R\|_{\mathrm{op}} \leq \beta$ and a strictly positive minimum Jacobian singular value $\sigma_{\min}(J_R) \geq m > 0$, are best examined per reconstruction head (Eq. 6), since Definition 1 guarantees that each column is decoded independently through its own linear head.

_Numerical columns._ The reconstruction map for numerical column $i$ is $\widehat{x}_i^{\mathrm{num}} = \widehat{e}_i^{\mathrm{num}} \widehat{w}_i^{\mathrm{num}} + \widehat{b}_i^{\mathrm{num}}$ (Eq. 6), which is affine in the decoded token $\widehat{e}_i^{\mathrm{num}} \in \mathbb{R}^{d_e}$. Because the map is affine, its Hessian vanishes identically, so the bounded-Hessian assumption is trivially satisfied with $\beta = 0$ (here $\beta$ denotes the Hessian bound of Proposition 1, not the KL weight of Eq. 7). The minimum-singular-value condition then reduces to requiring that the learned weight vector $\widehat{w}_i^{\mathrm{num}} \in \mathbb{R}^{d_e \times 1}$ is nonzero. This is not guaranteed architecturally, but is overwhelmingly likely under gradient-based training: a zero weight vector would produce a constant output for all inputs, incurring a large reconstruction penalty $\ell_{\mathrm{recon}}$ (Eq. 7).

_Categorical columns: bounded Hessian._ The reconstruction map for categorical column $j$ is $\widehat{p}_j^{\mathrm{cat}} = \mathrm{Softmax}(\widehat{e}_j^{\mathrm{cat}} \widehat{W}_j^{\mathrm{cat}} + \widehat{b}_j^{\mathrm{cat}})$ (Eq. 6), which composes a linear projection with a softmax nonlinearity. The softmax is $C^\infty$ everywhere, so the Hessian of the composed map is continuous and bounded on any bounded domain. Concretely, softmax outputs lie in $[0, 1]$ and the Hessian involves only products of these bounded quantities, so the bounded-curvature requirement $\|H_R\|_{\mathrm{op}} \leq \beta$ holds for a finite $\beta$.

_Categorical columns: minimum singular value._ The condition $\sigma_{\min}(J_R) > 0$ requires more care for categorical heads. The softmax maps into the simplex $\Delta^{C_j - 1}$, whose outputs sum to one; its Jacobian $\mathrm{diag}(p) - pp^\top$ therefore annihilates the all-ones direction $\mathbf{1}$ and is rank-deficient in the ambient $\mathbb{R}^{C_j}$. Strictly, $\sigma_{\min} > 0$ thus holds only when the output space is understood as the tangent space of the simplex $\Delta^{C_j - 1}$ (dimension $C_j - 1$) rather than full $\mathbb{R}^{C_j}$. This is the natural viewpoint here, since generated categorical probabilities are always projected onto the simplex before $\arg\max$ decoding (Eq. 6). Under this intrinsic-dimension view, the condition requires the composed Jacobian $(\mathrm{diag}(p) - pp^\top)(\widehat{W}_j^{\mathrm{cat}})^\top$ to have rank $C_j - 1$, i.e. the projection $\widehat{W}_j^{\mathrm{cat}} \in \mathbb{R}^{d_e \times C_j}$ does not collapse directions that the softmax Jacobian can still distinguish. This is likely under reconstruction-loss training (Eq. 7), but could in principle fail for near-degenerate categories that collapse to a simplex vertex: at a vertex ($p \approx e_k$) the softmax Jacobian $\mathrm{diag}(p) - pp^\top$ approaches zero, so its nontrivial singular values shrink exponentially with the logit gap, regardless of $\widehat{W}_j^{\mathrm{cat}}$.

_Empirical evidence._ We empirically do not observe such degeneracies on any of the twelve evaluation settings. On the six UCI benchmarks, TabFlowM achieves strong column-wise MLE scores (Table 3) and Shape fidelity (Table 1), indicating that the learned reconstruction heads maintain sufficient rank along the transport paths actually traversed by the flow. On the six million-scale BAF fraud variants (Table 7), where class imbalance exceeds 100:1, TabFlowM attains the best AUC-PR on five of six variants and the best Trend on all six. Because AUC-PR directly measures minority-class predictive structure, a categorical head that had collapsed

to a simplex vertex for the rare fraud label would map all generated tokens to the majority class and destroy the minority signal that AUC-PR measures; the consistently high AUC-PR therefore provides direct evidence that $\sigma_{\min}(J_R) > 0$ is maintained even for severely under-represented categories at scale.

Overall, we regard the regularity conditions of Proposition 1 as practically satisfied in the regimes explored. They are trivially met for numerical heads (affine map, zero Hessian) and naturally satisfied for categorical heads under the intrinsic simplex geometry, with the caveat that extreme categorical sparsity could in principle violate $\sigma_{\min} > 0$ near simplex vertices. We flag this as a direction for future theoretical refinement.

### A.4.2 Endpoint Control and Wasserstein Convergence of the Minimalist FM Surrogate (Proposition 2)

**Proposition 2 (Endpoint control from pathwise velocity regression)** *Under the linear-path FM surrogate, if the learned velocity field is Lipschitz in state, then small flow-matching regression error along the reference coupling path implies small endpoint error, and hence Wasserstein closeness between the generated token-space distribution and the target token-space distribution.*

**Proof.** Let $e_t := x_t - x_t^\star$. Since $x_0 = \varepsilon = x_0^\star$, we have $e_0 = 0$. Also,

$$\dot{e}_t = v_\theta(x_t, t) - v^\star = \big(v_\theta(x_t, t) - v_\theta(x_t^\star, t)\big) + \big(v_\theta(x_t^\star, t) - v^\star\big).$$

Hence, by the triangle inequality and the $L$-Lipschitz assumption,

$$\frac{d}{dt}\|e_t\|_2 \ \leq\ L\|e_t\|_2 + r_\theta(t; \varepsilon, \tilde{z}_0).$$

Applying Grönwall's inequality with $e_0 = 0$ yields

$$\|e_1\|_2 \leq \int_0^1 e^{L(1-s)} r_\theta(s; \varepsilon, \tilde{z}_0)\, ds \leq e^L \int_0^1 r_\theta(s; \varepsilon, \tilde{z}_0)\, ds.$$

Since $x_1^\star = \tilde{z}_0$, this proves

$$\|x_1 - \tilde{z}_0\|_2 \leq e^L \int_0^1 r_\theta(s; \varepsilon, \tilde{z}_0)\, ds.$$

Now couple $x_1$ and $\tilde{z}_0$ using the same draw $(\varepsilon, \tilde{z}_0)$. By the definition of $W_1$,

$$W_1(P_\theta, P_{\mathrm{tok}}) \leq \mathbb{E}\|x_1 - \tilde{z}_0\|_2 \leq e^L\, \mathbb{E}\left[\int_0^1 r_\theta(t; \varepsilon, \tilde{z}_0)\, dt\right].$$

Next, because $t \sim \mathrm{Unif}[0, 1]$, Eq. (15) implies

$$2L_{\mathrm{FM}}(\theta) = \mathbb{E}\left[\int_0^1 \|v_\theta(x_t^\star, t) - v^\star\|_2^2\, dt\right].$$

Using Cauchy–Schwarz on $[0, 1]$,

$$\int_0^1 r_\theta(t)\, dt \leq \left(\int_0^1 r_\theta(t)^2\, dt\right)^{1/2},$$

and then Jensen's inequality gives

$$\mathbb{E}\left[\int_0^1 r_\theta(t)\, dt\right] \leq \sqrt{\mathbb{E}\left[\int_0^1 r_\theta(t)^2\, dt\right]} = \sqrt{2L_{\mathrm{FM}}(\theta)}.$$

Therefore,

$$W_1(P_\theta, P_{\mathrm{tok}}) \leq e^L \sqrt{2L_{\mathrm{FM}}(\theta)}.$$

Finally, if $\widehat{x}_1$ is the numerical endpoint produced by the solver, then by the triangle inequality,

$$\|\widehat{x}_1 - \tilde{z}_0\|_2 \leq \|\widehat{x}_1 - x_1\|_2 + \|x_1 - \tilde{z}_0\|_2.$$

Taking expectations and infimizing over couplings yields

$$W_1(\widehat{P}_\theta, P_{\text{tok}}) \leq \mathbb{E}\|\widehat{x}_1 - x_1\|_2 + W_1(P_\theta, P_{\text{tok}}) \leq \eta_h + e^L \sqrt{2L_{\text{FM}}(\theta)}.$$

This completes the proof.

### A.5 Hyperparameter Search and Hardware Specifications

All experiments are conducted on AMD Threadripper PRO 7955WX (16 cores), an NVIDIA RTX 6000 Ada (48 GB), and 128 GB DDR5 RAM. For each baseline, we performed a grid search over hyperparameter values reported in prior literature specifically Zhang et al. (2024); Shi et al. (2024). Note that TabSyn and TabFlowM shares the same parameter configuration. The final hyperparameters used for benchmarking are summarized below.

Table 12: Hyperparameters for TABSYN & TABFLOWM.

| Parameter | Value |
|---|---|
| token_dim (d) | 4 |
| vae_num_layers (encoder/decoder) | 2 |
| vae_transformer_ffn_dim | 128 |
| beta_max | 0.01 |
| beta_max (Shoppers) | 0.001 |
| beta_min | 1e-5 |
| lambda | 0.7 |
| diffusion_mlp_hidden_dim | 1024 |

Table 13: Hyperparameters for TABDIFF (from the provided config file).

| Parameter | Value |
|---|---|
| data.dequant_dist | none |
| data.int_dequant_factor | 0 |
| diffusion_params.cat_scheduler | log_linear |
| diffusion_params.edm_params.net_conditioning | sigma |
| diffusion_params.edm_params.precond | True |
| diffusion_params.edm_params.sigma_data | 1 |
| diffusion_params.noise_dist | uniform_t |
| diffusion_params.noise_dist_params.P_mean | -1.2 |
| diffusion_params.noise_dist_params.P_std | 1.2 |
| diffusion_params.noise_schedule_params.eps_max | 0.001 |
| diffusion_params.noise_schedule_params.eps_min | 1e-05 |
| diffusion_params.noise_schedule_params.k_init | -6 |
| diffusion_params.noise_schedule_params.k_offset | 1 |
| diffusion_params.noise_schedule_params.rho | 7 |
| diffusion_params.noise_schedule_params.rho_init | 7 |
| diffusion_params.noise_schedule_params.rho_offset | 5 |
| diffusion_params.noise_schedule_params.sigma_max | 80 |
| diffusion_params.noise_schedule_params.sigma_min | 0.002 |
| diffusion_params.num_timesteps | 50 |
| diffusion_params.sampler_params.second_order_correction | True |
| diffusion_params.sampler_params.stochastic_sampler | True |
| diffusion_params.scheduler | power_mean |
| train.main.batch_size | 4096 |
| train.main.c_lambda | 1 |
| train.main.check_val_every | 500 |
| train.main.closs_weight_schedule | anneal |
| train.main.d_lambda | 1 |
| train.main.ema_decay | 0.997 |
| train.main.factor | 0.9 |
| train.main.lr | 0.001 |
| train.main.lr_scheduler | reduce_lr_on_plateau |
| train.main.reduce_lr_patience | 50 |
| train.main.steps | 500 |
| train.main.weight_decay | 0 |
| sample.batch_size | 10000 |
| unimodmlp_params.bias | True |
| unimodmlp_params.d_token | 4 |
| unimodmlp_params.dim_t | 1024 |
| unimodmlp_params.factor | 32 |
| unimodmlp_params.n_head | 1 |
| unimodmlp_params.num_layers | 2 |
| unimodmlp_params.use_mlp | True |

Table 14: Hyperparameters for TVAE.

| Parameter | Value |
|---|---|
| enforce_min_max_values | True |
| enforce_rounding | True |
| embedding_dim | 128 |
| compress_dims | (128, 128) |
| decompress_dims | (128, 128) |
| l2scale | 1e-05 |
| batch_size | 500 |
| verbose | True |
| epochs | 100 |
| loss_factor | 2 |
| cuda | True |

Table 15: Hyperparameters for CTGAN.

| Parameter | Value |
| --- | --- |
| enforce_min_max_values | True |
| enforce_rounding | True |
| locales | ['en_US'] |
| embedding_dim | 128 |
| generator_dim | (256, 256) |
| discriminator_dim | (256, 256) |
| generator_lr | 0.0002 |
| generator_decay | 1e-06 |
| discriminator_lr | 0.0002 |
| discriminator_decay | 1e-06 |
| batch_size | 500 |
| discriminator_steps | 1 |
| log_frequency | True |
| verbose | True |
| epochs | 100 |
| pac | 10 |
| cuda | True |

Table 16: Hyperparameters for CopulaGAN.

| Parameter | Value |
| --- | --- |
| enforce_min_max_values | True |
| enforce_rounding | True |
| locales | ['en_US'] |
| numerical_distributions | {} |
| default_distribution | beta |

