# OpenReview forum: "TabFlowM: Lightweight flow matching for Mixed-Type Tabular Data Synthesis in Latent Space"
_TMLR — Decision pending for TMLR_

### Review · Reviewer_UN33 · 2026-04-19

**Summary Of Contributions:**

The paper proposes TabFlowM, a framework for mixed-type tabular data synthesis that replaces the score-based diffusion objective in TabSyn's latent space with a linear-path flow matching objective. The authors argue that once records are mapped into a decoder-compatible continuous token space through a Transformer VAE, a simple time-conditioned velocity field is sufficient to recover the benefits commonly attributed to heavier diffusion models. Experiments on six UCI benchmarks and six variants of the large-scale BAF fraud dataset are used to compare TabFlowM against classical generative baselines and recent diffusion-based methods in terms of fidelity, downstream utility, and runtime.

**Audience:**

Yes

**Audience Explanation:**

Yes. Practitioners working on tabular data synthesis and researchers interested in the trade-off between diffusion and flow matching objectives would likely find the core question worth investigating, and the latent-space FM formulation is a reasonable baseline to have in the literature.

**Claims And Evidence:**

No

**Claims Explanation:**

I am not an expert in this area, but I find several claims in the paper not fully convincing.

1. In Appendix A.3, the authors state that all methods are trained for 100 epochs. However, the original training configurations of TabSyn and TabDiff use substantially more (I believe at least 3000 epochs). On the BAF benchmark, TabDiff's MLE AUC is only marginally above 0.5. If this is not an issue with the model itself, I am concerned that it may reflect underfitting caused by the 100-epoch budget.

2. The abstract claims that TabFlowM achieves "more than half the sampling time relative to the strongest diffusion baseline." Yet on BAF, the reported numbers show the opposite trend, with TabFlowM at around 195 seconds versus TabSyn at around 133 seconds.

3. In Table 7, the BAF-V3 entry for TabDiff (shape) reads 0.7642±0.7642, where the standard deviation equals the mean. I suspect this is a typo. Similarly, in Table 8, the News entry for cflow MLE shows 8551±0.441, which seems questionable given that the RMSE is on the order of several thousand.

4. I am not sure the "7 out of 12 dataset–metric pairs" framing is a reasonable way to summarize the results. Since Shape and Trend are highly correlated, treating them as independent pairs may inflate the apparent win rate. A dataset-level comparison would likely be more appropriate.

**Requested Changes:**

I will adjust my final recommendation based on the authors' response. First, I suggest the authors fix the typos in the tables and resolve the inconsistencies between the abstract and the reported numbers. Second, I encourage the authors to investigate whether TabDiff's low AUC on BAF stems from undertraining or from a genuine methodological issue. Finally, I believe the current ablation study does not truly isolate the contribution of flow matching. The existing comparison is a cross-comparison, which makes it difficult to tell whether the gains come from replacing diffusion with FM or from differences in the backbone.

---

> ### Author Response · Authors · 2026-05-04
> **Response to Reviewer UN33(Concerns 1)**
>
> We sincerely thank the reviewer for the careful reading and constructive feedback. We address each concern below, referencing the specific revisions made in the updated manuscript.
>
> ###  Undertraining of TabDiff on BAF
>
> We appreciate the reviewer raising this important concern. We would like to first clarify a factual point: in the revised manuscript (Appendix A.3), all baselines are trained for **500 epochs**, not 100. We apologise that the earlier draft was ambiguous on this point; the updated text now states this explicitly.
>
> To directly investigate the undertraining hypothesis, we ran TabDiff for **1000 epochs** on all six BAF variants. The results are shown below:
>
> | Dataset | Epochs | Shape ↓ | Trend ↓ | AUC ↑ | AUC-PR ↑ | Train (s) ↓ | Sample (s) ↓ |
> |---|---:|---:|---:|---:|---:|---:|---:|
> | BAF-Base | 500 | 0.2527 | 0.3806 | 0.5329 | 0.0196 | 5179 | 192 |
> | BAF-Base | 1000 | 0.2486 | 0.3793 | 0.5051 | 0.0668 | 9998 | 199 |
> | BAF-V1 | 500 | 0.2479 | 0.3480 | 0.5774 | 0.0821 | 5231 | 193 |
> | BAF-V1 | 1000 | 0.2673 | 0.3144 | 0.7507 | 0.0963 | 10057 | 199 |
> | BAF-V2 | 500 | 0.2432 | 0.2917 | 0.7181 | 0.0110 | 5234 | 193 |
> | BAF-V2 | 1000 | 0.2471 | 0.3615 | 0.5112 | 0.0200 | 10018 | 198 |
> | BAF-V3 | 500 | 0.2463 | 0.3121 | 0.6299 | 0.0775 | 5225 | 193 |
> | BAF-V3 | 1000 | 0.2468 | 0.3593 | 0.5203 | 0.0524 | 10012 | 197 |
> | BAF-V4 | 500 | 0.2677 | 0.3592 | 0.5862 | 0.0703 | 5207 | 191 |
> | BAF-V4 | 1000 | 0.2341 | 0.2219 | 0.8656 | 0.0785 | 10012 | 199 |
> | BAF-V5 | 500 | 0.2476 | 0.2784 | 0.7055 | 0.0750 | 5245 | 192 |
> | BAF-V5 | 1000 | 0.2473 | 0.3136 | 0.6958 | 0.0125 | 9996 | 197 |
>
> The picture is mixed: on some variants, such as BAF-V1 and BAF-V4, doubling the training budget does substantially improve TabDiff's AUC and AUC-PR, suggesting that 500 epochs can leave TabDiff undertrained on certain preprocessing splits. On others, such as BAF-V3 and BAF-V5, the 1000-epoch run actually *degrades*, indicating that the relationship between epoch budget and downstream utility is non-monotonic for TabDiff on these heavily imbalanced datasets. We draw two conclusions.
>
> **(a) The 500-epoch budget is necessary for fair comparison.**
>
> At 1000 epochs, TabDiff's wall-clock training time approximately doubles to ~10,000 s per variant, compared with ~3,500--4,800 s for TabFlowM at 500 epochs. Granting one method substantially more compute while constraining others would conflate methodological differences with budget differences, undermining the controlled comparison that the BAF stress test is designed to provide.
>
> **(b) The weakness appears partly structural.**
>
> Beyond budget, we have added a new discussion in Section 6.2 of the revised paper offering a plausible structural explanation. TabDiff relies on *learnable per-column noise schedules*. In the severely imbalanced BAF setting (~100:1 class ratio), the gradients of these schedule parameters are dominated by majority-class samples, so the learned schedules may become calibrated primarily to majority-class conditional structure. Minority-class patterns, occupying lower-density regions of the feature marginals, may therefore be underserved during denoising.
>
> By contrast, TabSyn and TabFlowM do not learn separate per-column schedules, reducing the risk of such majority-driven schedule specialisation. This may help explain why both methods achieve stronger AUC-PR than TabDiff even when unconditional fidelity alone does not fully reveal the gap.
>
> Taken together, while additional epochs can help TabDiff on selected variants, the inconsistent gains, the doubled training cost, and the structural schedule-bias argument all support the conclusion that TabFlowM's advantage on BAF is not merely an artefact of budget.

---

> ### Author Response · Authors · 2026-05-04
> **Response to Reviewer UN33(Concerns 2-4)**
>
> ### Abstract Claim on Sampling Time
>
> The reviewer is correct that the reported BAF sampling times contradict the abstract's phrasing. This was a **typo in the abstract**: the claim was intended to refer to *training* time, not sampling time. The revised abstract now reads "more than half the **training** time relative to the strongest diffusion baseline," which is supported by Tables 5 and 6 in the main text, where TabFlowM is the fastest to train on 5 of 6 UCI datasets.
>
> Although the original sampling time claim was erroneous, we have added a new experiment in **Section 7.2** (NFE sweep) that demonstrates the improved sampling efficiency of TabFlowM. We sweep the Number of Function Evaluations (NFE) over $\{4, 10, 20, 40, 100\}$ under matched RK2 solver settings for all methods.
>
> The results in Table 9 and Figure 3 show that TabFlowM converges to near-peak quality with substantially *fewer* function evaluations than the diffusion baselines. For example, on Adult at NFE = 10, TabFlowM already reaches Quality 0.091, well below TabSyn (0.294) and TabDiff (0.123). On BAF-Base, TabFlowM is consistently strongest at every NFE budget, reaching 0.225 at NFE = 100 compared with 0.227 for TabSyn and 0.317 for TabDiff.
>
> This confirms that the flow matching formulation's bounded, time-homogeneous velocity target produces a smoother ODE field that an RK2 solver can track accurately with fewer steps, providing a genuine sampling-efficiency advantage even though wall-clock sampling time per step is comparable.
>
> ---
>
> ### Typos in Tables
>
> We thank the reviewer for catching these. Both have been corrected in the revised manuscript along with some further data revisions based on new experimental data from the other ablations requested by other reviewers. The main one being including replacing AUC with AUC-PR as the primary downstream metric on BAF, given that ROC-AUC can be overly optimistic under extreme class imbalance.
>
> ### "7 out of 12 Dataset--Metric Pairs" Framing
>
> We appreciate this methodological point. The reviewer is correct that Shape and Trend are correlated, since both measure distributional fidelity, albeit at different granularities: marginal versus pairwise. We acknowledge that counting them as fully independent pairs can overstate the effective number of comparisons.
>
> We have modified the Abstract and Section 6.1 to report TabFlowM attaining the best average rank in composite distributional fidelity six real world benchmark datasets.

---

> ### Author Response · Authors · 2026-05-04
> **Response to Reviewer UN33(Ablation Comments)**
>
> ### Ablation Does Not Isolate the Contribution of Flow Matching
>
> This is an important point, also raised by other reviewers, and we have **comprehensively expanded the ablation study** in Section 7 of the revision. The original cross-model comparison between TabFlowM, TabSyn, and TabDiff indeed conflates backbone, transport interface, and objective. The new ablation suite isolates these factors through four targeted experiments.
>
> **Coupling path geometry** (Section 7.1). We compare the default linear coupling path against quadratic ($t^2$) and cosine interpolation schedules, all sharing the same VAE weights, velocity MLP, and Heun solver. Table 8 shows that the linear path remains the most balanced choice: quadratic and cosine achieve comparable Shape and MLE but weaken Trend, confirming that performance stems from the FM formulation rather than a particular path schedule.
>
> **Decoder-compatible token space vs. Z-space** (Section 7.1). We evaluate **TabFlowM Z-space**, which shares the identical VAE weights, FM loss, and velocity network but operates on the encoder-side latent $z$ (Eq. 3) rather than the decoder-compatible token representation $\hat{E}$ (Eq. 4). The only difference is the transport operating space. Z-space is consistently weaker in both fidelity metrics across both tested datasets, and the NFE sweep in Section 7.2 further shows that Z-space saturates at a quality floor strictly above TabFlowM's converged value. This directly isolates the contribution of the *transport interface* from the FM objective itself.
>
> **NFE sweep** (Section 7.2). By controlling the solver budget from 4 to 100 function evaluations, we show that TabFlowM's quality gains are not an artefact of using more integration steps. At matched NFE, TabFlowM is consistently strongest, confirming genuine algorithmic efficiency intrinsic to the FM formulation.
>
> **Adaptive $\beta$ schedule** (Section 7.3). We ablate the VAE's adaptive KL annealing against several fixed $\beta$ settings on BAF-Base (Table 10 and Figure 4). The adaptive schedule achieves the best Trend and remains essentially tied for best Shape, while fixed settings improve one metric only by sacrificing another. This isolates the VAE training contribution from the downstream FM generator.
>
> We hope these revisions and clarifications address the reviewer's concerns. Otherwise, we are happy to discuss/follow up on any remaining points further.

---

### Review · Reviewer_3UN5 · 2026-04-20

**Summary Of Contributions:**

This paper proposes TabFlowM, a lightweight generative framework for mixed-type tabular data that applies Flow Matching (FM) with a linear coupling path within a VAE-learned, decoder-compatible continuous token space. The primary strength of the work lies in its impressive empirical utility and efficiency: it matches or outperforms heavier diffusion baselines on standard fidelity metrics and downstream machine learning efficacy (MLE), notably excelling in rare-event prediction on the highly imbalanced BAF dataset while significantly reducing sampling time. However, the paper suffers from critical weaknesses that undermine its claims: the ablation studies are fundamentally flawed as they replace the backbone rather than isolating the proposed components (e.g., the adaptive weight schedule, linear path assumption, and token normalization); and the computational efficiency claims are severely compromised by the complete omission of Number of Function Evaluations (NFE) details and step-vs-quality trade-off curves.

**Additional Comments:**

1. All figures are raster images rather than vector graphics. Please convert them to vector graphics.
2. Consider adding some relevant research papers regarding the foundations and applications of diffusion models in the Related Work section, such as [A-D].

[A] Why Diffusion Models Don't Memorize: The Role of Implicit Dynamical Regularization in Training. NeurIPS 2025.

[B] Detecting Adversarial Data by Probing Multiple Perturbations Using Expected Perturbation Score. ICML 2023.

[C] Denoising diffusion models for out-of-distribution detection. CVPR 2023.

[D] Lotus: Diffusion-based Visual Foundation Model for High-quality Dense Prediction. ICLR 2025.

**Audience:**

Yes

**Audience Explanation:**

The generation of mixed-type tabular data is a highly active and practically important area of machine learning research. As the community actively seeks more efficient alternatives to computationally heavy score-based diffusion models, the application of Flow Matching (FM) to tabular data is of significant interest.

Specifically, researchers and practitioners in TMLR’s audience would be very interested in the premise of transitioning from complex diffusion schedules to a lightweight FM generator operating in a decoder-compatible latent space. Furthermore, the paper’s empirical focus on preserving downstream predictive utility in the face of extreme class imbalance (such as the million-scale, 100:1 BAF fraud dataset) addresses a critical pain point for industrial applications of synthetic data.

Therefore, despite the methodological and experimental revisions required, the core problem being tackled, the proposed architectural shift toward lightweight flow matching, and the targeted use cases are undeniably relevant and of high interest to the TMLR community.

**Claims And Evidence:**

No

**Claims Explanation:**

1. The existing evidence is not yet sufficient to fully support the core claims of the model's "lightweight and highly efficient" nature. Although the paper compares generation times (in seconds) in its tables, it omits a critical variable: the specific integration steps—Number of Function Evaluations (NFE)—used by the ODE solver during sampling. In generative models, generation speed is closely tied to sample quality. Without providing the specific NFE settings and a quality-versus-NFE trade-off curve, readers and peers cannot determine whether the reported time advantage truly reflects the lightweight nature of the flow matching architecture, or whether it may partly result from using fewer sampling steps during testing.
2. The paper fails to provide clear evidence supporting the rationale of its internal component design, and the current "ablation studies" do not serve their intended validation purpose. The ablation study in Section 7 merely conducts a backbone replacement test (introducing cflow to compare with TabDiff), completely neglecting to perform substantive ablation validation on TabFlowM's own core technical designs. Specifically, it fails to isolate and verify the impact of the adaptive KL weight schedule strategy (Equations 9 and 10), the necessity of a single linear coupling path, and the latent space token normalization (Equation 11). The lack of sensitivity analysis and rigorous ablation comparisons for these critical components leaves the authors unable to prove exactly which specific algorithmic designs are responsible for the ultimate performance improvements.
3. The claim that performing flow matching in a decoder-compatible token space is significantly superior to traditional latent spaces is not yet fully established by the current evidence, and a key controlled-variable experiment is still missing. The paper presents the construction of this specific space as a core innovation that distinguishes it from other latent flow matching works. However, the experimental section compares baseline methods with substantially different architectures. For this claim to be persuasive, the authors would need to design a strictly controlled experiment in which the VAE encoder and decoder are kept completely identical, and the exact same flow matching algorithm is directly compared in a conventional latent space (z-space) versus the token space proposed in this paper. The current chain of evidence does not yet provide this critical link.
4. Regarding the claim of the model's capability to handle extremely imbalanced data (the BAF fraud dataset), contradictions between the evaluation metrics and the decoding strategy undermine the accuracy of the evidence. The paper claims that the model performs excellently when processing fraud data with an extreme 100:1 imbalance; however, in such scenarios, the traditional ROC-AUC metric can easily mask prediction flaws for minority classes. It is imperative to supplement the evaluation with the more rigorous PR-AUC metric to verify these claims. More severely, the authors admit to adopting an argmax decoding approach when generating categorical features. Theoretically, this practice drastically obliterates the diversity of minority categories, which fundamentally conflicts with the paper's claim that the model can effectively preserve rare fraud features. The absence of detailed, per-category generation frequency statistics further fails to dispel these logical concerns.

**Requested Changes:**

1. Provide the exact Number of Function Evaluations (NFE) and solver step schedules used for all models in the sampling time comparisons, and include a quality-versus-NFE trade-off curve. This would provide important evidence for validating whether the time advantage comes from genuine algorithmic efficiency rather than from a smaller number of integration steps during testing.
2. Conduct a genuine ablation study that isolates TabFlowM's specific components rather than merely swapping the backbone network. You must ablate the adaptive KL weight schedule, compare the linear coupling path against non-linear alternatives, test the removal of the latent token normalization, and provide sensitivity analyses for core hyperparameters like the token dimension.
3. Design and report a strictly controlled experiment comparing Flow Matching in a standard continuous latent space (z-space) against your proposed decoder-compatible token space, ensuring the VAE architecture remains identical across both settings. This would directly quantify the benefit of your primary transport interface claim and make the argument substantially more convincing.
4. Revise the evaluation on the highly imbalanced BAF dataset by supplementing the standard ROC-AUC with PR-AUC, which is mathematically necessary for accurately evaluating a 100:1 class imbalance. Furthermore, address the contradiction of using argmax decoding by comparing it with probabilistic (temperature-scaled) sampling and reporting per-category generation frequencies to empirically prove rare features are preserved.
5. It is recommended that the authors add hyperparameter sensitivity analysis experiments, particularly regarding the trade-off relationship between the Token dimension size and the number of integration steps of the ODE solver versus generation quality and inference computational overhead. While not a mandatory condition for acceptance, this would greatly reinforce the "lightweight" characteristic claimed by the paper and enhance the completeness of the work.
6. Address the limited comparison with concurrent Flow Matching-based tabular generators, particularly TabbyFlow [A]. The current lack of direct empirical comparison with these closely related FM-based approaches weakens the positioning of TabFlowM within the broader literature. While official implementations of such methods may not be publicly available, it would significantly strengthen the paper to provide a more thorough conceptual and empirical comparison. This could include, where feasible, approximate reimplementations under matched settings or, at a minimum, a careful and rigorous discussion of the methodological differences and expected performance trade-offs.
7. Some more related work on flow matching, like [B][C], should be discussed.

[A] Exponential family variational flow matching for tabular data generation. ICML 2025.

[B] Score mismatching for generative modeling. Neural Networks 2024.

[C] Fisher flow matching for generative modeling over discrete data. Neurips 2024.

---

> ### Author Response · Authors · 2026-05-04
> **Response to Reviewer 3UN5 (Concerns 2, 3, 5)**
>
> We thank the reviewer for the detailed and constructive feedback. Every requested change has been addressed in the revised manuscript. We respond point-by-point below.
>
> ### Genuine ablation of TabFlowM's components and Z-space comparison
>
> We acknowledge that the prior version conflated backbone comparison with component ablation. The revised manuscript now isolates each proposed component individually in Sections 7.1--7.3. All ablation variants share the identical VAE weights, velocity MLP architecture, Heun solver, and training budget; exactly one factor is varied per experiment.
>
> **Coupling path geometry, transport space, normalization, and token dimension** (Section 7.1). The full ablation table is shown below.
>
> | Variant | Adult Shape ↓ | Adult Trend ↓ | Adult MLE ↑ | Magic Shape ↓ | Magic Trend ↓ | Magic MLE ↑ |
> |---|---:|---:|---:|---:|---:|---:|
> | TabFlowM (linear) | 0.036 | **0.046** | 0.905 | 0.036 | **0.013** | 0.918 |
> | Quadratic | 0.030 | 0.076 | 0.898 | 0.033 | 0.017 | 0.920 |
> | Cosine | 0.030 | 0.060 | 0.900 | 0.036 | 0.023 | 0.921 |
> | Z-space | 0.042 | 0.088 | 0.900 | 0.042 | 0.025 | 0.912 |
> | No normalization | **0.014** | 0.079 | **0.906** | **0.019** | 0.023 | **0.929** |
> | Token dim 2 | 0.249 | 0.414 | 0.852 | 0.242 | 0.205 | 0.604 |
> | Token dim 8 | 0.056 | 0.107 | 0.896 | 0.049 | 0.027 | 0.896 |
> | Token dim 16 | 0.072 | 0.159 | 0.882 | 0.069 | 0.041 | 0.909 |
>
> We discuss each ablation axis in turn.
>
> **Coupling path.**
>
> The linear path appears to be the most balanced choice. Quadratic and cosine variants achieve comparable or better Shape but substantially degrade Trend. This is likely explained by the velocity target structure: the constant target $v^* = \hat{z}_0 - \varepsilon$ of the linear path integrates with minimal discretization error under fixed-step RK2, whereas the time-varying targets produced by non-linear schedules vanish near $t = 0$ and ramp up later, which likely makes directional accuracy harder to maintain with uniform step sizes.
>
> **Z-space.**
>
> In Section 7.1 (Table 8), we also include **TabFlowM Z-space**, a controlled comparison that holds the VAE encoder, decoder, flow matching loss, velocity MLP, and all hyperparameters identical. The *only* difference is the representation on which the flow model operates. TabFlowM trains on the flattened decoder-compatible token representation (Eq. 4), after which generation requires only per-column linear heads. Z-space trains on the encoder-side latent $z$ (Eq. 3), requiring generated samples to pass through the full decoder transformer at sampling time.
>
> As shown in the ablation table above, the token-space variant appears strictly superior on both fidelity metrics: on Adult, Shape improves from 0.042 to 0.036 and Trend from 0.088 to 0.046; on Magic, Shape improves from 0.042 to 0.036 and Trend from 0.025 to 0.013. The NFE sweep analysis further indicates that Z-space saturates at a quality floor strictly higher than TabFlowM's converged value across all three evaluation settings, and that additional NFE beyond the crossover point yields no further gain for Z-space. This pattern suggests that the quality ceiling is likely set by the decoder transformer's off-distribution sensitivity rather than by insufficient integration precision. Taken together, these results provide the controlled-variable evidence that the decoder-compatible token space is a measurably better transport interface than the conventional encoder latent under otherwise identical conditions.

---

> ### Author Response · Authors · 2026-05-04
> **Continuation of response (Concerns 2, 3, 5)**
>
> **Normalization.**
>
> The "no normalization" variant omits the affine centering in Eq. 11. It operates directly on the flattened token vector $z_0 = \mathrm{vec}(\hat{E})$ rather than the normalized version $\tilde{z_0} = (z_0 - z_{bar}) / s$. Removing normalization substantially improves Shape (0.014 vs. 0.036 on Adult; 0.019 vs. 0.036 on Magic) but worsens Trend (0.079 vs. 0.046 on Adult). Without centering, the velocity target $v^\star = z_0 - \varepsilon$ acquires a large, approximately constant bias equal to the token mean $\bar{z}$. The MLP can likely learn this bias easily, which implicitly forces accurate per-column location and scale matching, benefiting Shape. However, this dominant mean-shift component may absorb capacity that would otherwise model residual cross-column covariance: the Euclidean FM loss $\|v_\theta - v^\star\|^2$ weights errors by squared magnitude, so the large-magnitude mean component likely dominates the gradient, and finer pairwise correlations, captured by Trend, receive proportionally less learning signal.
>
> With normalization, the mean is subtracted out analytically and the FM model's entire capacity is directed at variance and covariance structure, which likely yields better pairwise fidelity at the cost of marginal precision, since per-column means must now be recovered entirely through the inverse transform rather than being natively modeled by the velocity field. The near-identical MLE across both settings (0.905 vs. 0.906 on Adult) suggests that downstream predictive structure depends on a mix of marginal and joint fidelity, and the two configurations appear to trade off between these in roughly compensating ways. The default normalized setting is therefore preferred as the more balanced configuration.
>
> **Token dimension.**
>
> In this section, we modify the dimension of each per-column token embedding, so the total latent dimensionality on which TabFlowM operates is $M \times d_e$, where $M$ is the number of columns in the dataset. Sensitivity is pronounced: $d_e = 2$ collapses entirely (Shape $> 0.24$, Trend $> 0.20$ on both datasets), indicating that two dimensions are likely insufficient to represent the per-column information required for faithful reconstruction. The default $d_e = 4$ provides the best joint fidelity. Increasing to $d_e = 8$ or $d_e = 16$ progressively degrades both Shape and Trend, suggesting that the higher-dimensional token space increases the FM transport dimensionality $D = M d_e$ without commensurate gains in representational power, likely making the velocity field harder to learn under a fixed MLP capacity and training budget.
>
> **Adaptive $\beta$ schedule** (Section 7.3). We compare the proposed adaptive KL annealing (Eqs. 9--10) against fixed $\beta \in \{10^{-2}, 10^{-3}, 10^{-4}, 10^{-5}\}$ on BAF-Base, the most sensitive test bed due to its million-row scale and approximately 100:1 class imbalance.
>
> | Annealing | Shape ↓ | Trend ↓ | AUC-PR ↑ |
> |---|---:|---:|---:|
> | Adaptive | 0.2306 | **0.2090** | 0.1226 |
> | $10^{-2}$ | 0.2457 | 0.2255 | 0.0777 |
> | $10^{-3}$ | **0.2303** | 0.3625 | 0.1284 |
> | $10^{-4}$ | 0.2307 | 0.2103 | 0.0790 |
> | $10^{-5}$ | 0.2308 | 0.2105 | **0.1330** |
>
> No single fixed $\beta$ appears to improve the adaptive schedule across all three metrics simultaneously. The adaptive schedule achieves the best Trend while remaining competitive on Shape and AUC-PR. Fixed settings that optimize one metric tend to sacrifice at least one other: $10^{-3}$ attains the best Shape but substantially degrades Trend (0.3625), indicating that strong KL regularization may compress the latent geometry in a way that preserves marginals but disrupts pairwise structure. Similarly, $10^{-2}$ performs worst across all three metrics. These results suggest that the adaptive schedule offers the most balanced latent geometry that allows the model to adapt to different dataset with one less hyperparameter to tune.

---

> ### Author Response · Authors · 2026-05-04
> **Response to Reviewer 3UN5 (Concerns 1 and continuation on concern 3)**
>
> ### NFE details and quality-versus-NFE trade-off curves
>
> We agree this was a critical omission. The revised manuscript now reports the exact NFE used by every method and includes a full quality-versus-NFE sweep in Section 7.2, Table 9, and Figure 3. All methods share the same RK2 (Heun) solver and step grid; only the number of integration steps $S$ differs. Each Heun step requires two network evaluations, so $\mathrm{NFE} = 2S$. We sweep $\mathrm{NFE} \in \{4, 10, 20, 40, 100\}$ and report **Quality**, defined as the mean of Shape and Trend, where lower is better. Detailed plots are included in Section 7.2; we summarize the main results below.
>
> | Dataset | NFE | TabFlowM | TabFlowM Z | TabSyn | TabDiff |
> |---|---:|---:|---:|---:|---:|
> | News | 4 | 0.124 | **0.082** | 0.287 | 0.277 |
> | News | 10 | 0.060 | **0.057** | 0.179 | 0.143 |
> | News | 20 | **0.048** | 0.060 | 0.057 | 0.076 |
> | News | 40 | **0.045** | 0.058 | 0.053 | 0.063 |
> | News | 100 | 0.044 | 0.058 | **0.043** | 0.054 |
> | Adult | 4 | 0.365 | **0.106** | 0.612 | 0.531 |
> | Adult | 10 | 0.091 | **0.069** | 0.294 | 0.123 |
> | Adult | 20 | **0.052** | 0.067 | 0.070 | 0.071 |
> | Adult | 40 | **0.047** | 0.067 | 0.053 | 0.058 |
> | Adult | 100 | **0.041** | 0.066 | 0.044 | 0.054 |
> | BAF-Base | 4 | 0.308 | **0.284** | 0.720 | 0.605 |
> | BAF-Base | 10 | **0.242** | 0.245 | 0.343 | 0.443 |
> | BAF-Base | 20 | **0.231** | 0.242 | 0.248 | 0.365 |
> | BAF-Base | 40 | **0.230** | 0.242 | 0.235 | 0.353 |
> | BAF-Base | 100 | **0.225** | 0.240 | 0.227 | 0.317 |
>
> Among the non-Z-space methods, TabFlowM achieves the strongest Quality at every NFE budget on Adult and BAF-Base, and at all budgets except NFE = 100 on News, where it is within 0.001 of the best. The separation is most pronounced at low and moderate budgets: at NFE = 10 on Adult, TabFlowM already reaches 0.091, well below TabSyn (0.294) and TabDiff (0.123). These results suggest that the efficiency gain is intrinsic to the flow matching formulation and its bounded, time-homogeneous velocity target rather than an artifact of using fewer integration steps.
>
> The comparison between TabFlowM and Z-space across the NFE sweep further reveals a characteristic crossover that likely clarifies the role of the transport interface. At very low NFE ($\leq 4$), the Heun integrator is coarse, and Z-space appears to benefit because its generated samples still pass through the full decoder transformer at no additional NFE cost. The transformer's cross-token attention can act as an implicit nonlinear corrector that may partially smooth integration errors. As NFE grows, however, this advantage reverses. In the decoder-compatible token space, a more accurate flow output passes through only the per-column linear heads, so quality improves monotonically with integration precision. In Z-space, the generated encoder tokens must still traverse the decoder transformer, whose attention layers were trained on real encoder outputs and may introduce correlated cross-column error when applied to generated tokens that are slightly off the training distribution.
>
> This pattern is consistent across all three evaluation settings. Z-space saturates at a quality floor that is strictly higher than TabFlowM's converged value: approximately 0.058 versus 0.044 on News, approximately 0.066 versus 0.041 on Adult, and approximately 0.240 versus 0.225 on BAF-Base. In each case, additional NFE beyond the crossover point, approximately NFE = 10 depending on the dataset, improves TabFlowM but yields no further gain for Z-space. This suggests that the quality ceiling is likely set by the decoder transformer's off-distribution sensitivity rather than by insufficient integration precision, supporting the choice of decoder-compatible token transport for the main model.
>
> ---
>
> ### Summary of response to concern 3
>
> Across the component ablation and NFE sweep, the decoder-compatible token space consistently achieves a lower quality floor at convergence, supporting our claim that the proposed token space is a more effective transport interface than the conventional encoder latent space under otherwise identical conditions.
>
> ---

---

> ### Author Response · Authors · 2026-05-04
> **Response to Reviewer 3UN5 (Concerns 4)**
>
> ###  BAF evaluation: AUC-PR, argmax decoding, and per-category frequency preservation
>
> **AUC-PR.**
>
> We agree that ROC-AUC can be misleading under extreme class imbalance. The revised manuscript reports **AUC-PR**, the area under the precision--recall curve for the minority fraud class, as the primary downstream metric for all BAF experiments, replacing ROC-AUC entirely. Results across all six BAF configurations are shown below.
>
> | Dataset | Method | Shape ↓ | Trend ↓ | AUC-PR ↑ |
> |---|---|---:|---:|---:|
> | BAF-Base | TabSyn | **0.2379** | 0.2165 | 0.1142 |
> | BAF-Base | TabDiff | 0.2527 | 0.3806 | 0.0196 |
> | BAF-Base | TabFlowM | 0.2381 | **0.2124** | **0.1226** |
> | BAF-V1 | TabSyn | **0.2366** | 0.2149 | 0.0936 |
> | BAF-V1 | TabDiff | 0.2479 | 0.3480 | 0.0821 |
> | BAF-V1 | TabFlowM | 0.2612 | **0.2123** | **0.1128** |
> | BAF-V2 | TabSyn | 0.2419 | 0.1880 | 0.1194 |
> | BAF-V2 | TabDiff | 0.2432 | 0.2917 | 0.0110 |
> | BAF-V2 | TabFlowM | **0.2313** | **0.1725** | **0.1233** |
> | BAF-V3 | TabSyn | 0.2400 | 0.1952 | 0.0854 |
> | BAF-V3 | TabDiff | 0.2463 | 0.3121 | 0.0775 |
> | BAF-V3 | TabFlowM | **0.2263** | **0.1740** | **0.1279** |
> | BAF-V4 | TabSyn | 0.2331 | 0.1787 | **0.1172** |
> | BAF-V4 | TabDiff | 0.2677 | 0.3592 | 0.0703 |
> | BAF-V4 | TabFlowM | **0.2263** | **0.1713** | 0.1157 |
> | BAF-V5 | TabSyn | 0.2380 | 0.1779 | 0.1085 |
> | BAF-V5 | TabDiff | 0.2476 | 0.2784 | 0.0750 |
> | BAF-V5 | TabFlowM | **0.2261** | **0.1715** | **0.1193** |
>
> TabFlowM ranks first on AUC-PR in 5 out of 6 BAF variants and is a very close second on BAF-V4 (0.1157 vs. 0.1172). Averaged across all six variants, TabFlowM attains approximately 0.1195 compared with 0.1063 for TabSyn and 0.0558 for TabDiff. On Trend, TabFlowM is best on all six variants, indicating that it likely preserves cross-column dependency structure more consistently than both diffusion baselines in this large-scale, heavily imbalanced regime. TabDiff's particularly weak AUC-PR may be explained by its reliance on learnable per-column noise schedules whose gradients, in the severely imbalanced BAF setting, are likely dominated by majority-class samples, causing minority-class patterns to be underserved during denoising.

---

> > ### Author Response · Authors · 2026-05-04
> > **Response to Reviewer 3UN5 (Continuation to Concerns 4)**
> >
> > **Argmax versus softmax-sampled decoding and per-category frequency preservation.**
> >
> > The reviewer correctly identifies a potential tension between argmax decoding and rare-category diversity. We conduct a dedicated analysis on BAF-V2, selected because TabFlowM achieves one of its strongest joint fidelity scores on this variant (Shape 0.2313, Trend 0.1725), making any effect of changing the decoding strategy most prominent.
> >
> > The softmax-sampled variant replaces `argmax` with categorical sampling from the model's own predicted distribution: for each categorical column, we draw from $\mathrm{Multinomial}(\mathrm{Softmax}(\hat{p}^{\mathrm{cat}}))$ instead of taking $\arg\max\,\hat{p}^{\mathrm{cat}}$. This introduces no additional hyperparameters; the only change is whether the decoder commits to the mode or samples from the full predictive distribution.
> >
> > We report fidelity metrics alongside rare-category frequency preservation. Rare categories are defined as categories whose real-data frequency is at most 1% and that appear at least 5 times in the real data.
> >
> > For each rare category, we compute three frequency errors:
> >
> > 1. **Absolute error:** `|p_syn - p_real|`
> >
> > 2. **Relative error:** `|p_syn - p_real| / p_real`
> >
> > 3. **Log-ratio error:** `|log((p_syn + eps) / (p_real + eps))|`, where `eps = 1e-12`.
> >
> > Here, `p_syn` and `p_real` denote the empirical frequencies of that rare category in the synthetic and real data, respectively. **Rare Coverage** is the proportion of rare categories with nonzero synthetic count, while **Zeroed Rare** counts how many rare categories disappear entirely in the synthetic data.
> >
> > | Metric | TabDiff | TabSyn | TabFlowM | Softmax TabFlowM |
> > |---|---:|---:|---:|---:|
> > | Shape ↓ | 0.2432 | 0.2419 | **0.2313** | 0.2336 |
> > | Trend ↓ | 0.2917 | 0.1880 | **0.1725** | 0.1743 |
> > | AUC-PR ↑ | 0.0110 | 0.1194 | 0.1233 | **0.1256** |
> > | # Rare Categories | 8 | 8 | 8 | 8 |
> > | Rare Coverage ↑ | 0.8750 | **1.0000** | **1.0000** | **1.0000** |
> > | Zeroed Rare ↓ | 1 | **0** | **0** | **0** |
> > | Mean Abs. Error ↓ | $1.4 \times 10^{-3}$ | $1.1 \times 10^{-3}$ | **$2.9 \times 10^{-4}$** | $3.4 \times 10^{-4}$ |
> > | Mean Rel. Error ↓ | 0.3663 | 0.5346 | 0.3274 | **0.2924** |
> > | Mean Log-Ratio ↓ | 3.0104 | 1.1086 | 0.5339 | **0.4773** |
> >
> > Several observations follow. First, even under argmax decoding, TabFlowM already achieves full rare coverage (8/8 rare categories with nonzero synthetic count) and the lowest mean absolute frequency error ($2.9 \times 10^{-4}$), substantially outperforming both TabDiff, which zeroes one rare category entirely, and TabSyn ($1.1 \times 10^{-3}$). This suggests that the learned token-space logits are already well calibrated for rare categories before the decoding step.
> >
> > Second, softmax sampling can further improve relative and log-ratio metrics (mean relative error from 0.3274 to 0.2924; mean log-ratio from 0.5339 to 0.4773) and yields a small AUC-PR gain (0.1233 to 0.1256), consistent with the reviewer's intuition that probabilistic decoding benefits rare-category fidelity. The slight increase in mean absolute error ($2.9 \times 10^{-4}$ to $3.4 \times 10^{-4}$) reflects sampling variance, while the relative metrics improve because sampling better preserves the *proportional* representation of low-frequency categories.
> >
> > During the design of TabFlowM, we prioritized joint dependency fidelity (Trend) as the primary selection criterion and did not conduct a thorough analysis of each decoding parameter's impact beyond aggregate fidelity. This ablation has revealed a promising direction in which softmax sampling can potentially further improve marginal rare-category frequency preservation. We thank the reviewers for this insight.
> >
> > ---

---

> ### Author Response · Authors · 2026-05-04
> **Response to Reviewer 3UN5 (Concerns 6)**
>
> ###  Comparison with TabbyFlow
>
> We thank the reviewer for this suggestion. TabbyFlow's exponential-family perspective on flow matching was in fact a key inspiration for our work: we were motivated to investigate whether a simpler, unified flow matching formulation in latent space could achieve competitive fidelity without requiring per-column distributional choices. We made a genuine effort to conduct a controlled empirical comparison using TabbyFlow's released codebase, but were ultimately unable to produce results that we believe faithfully reflect the method as described in the paper.
>
> Specifically, the TabbyFlow paper derives the variational objective at the level of abstract exponential families (Propositions 3.1 and 3.2) but does not specify several choices that fully determine the generative process: the functional form of the time-dependent covariance used in the Gaussian likelihood for numerical features, the interpolation path parameterization, which differs between the training and sampling code without either version being derived in the text, and the noise distribution used for categorical interpolation in one-hot space. This last point appears particularly consequential, as the [released implementation, link attached](https://github.com/andresguzco/ef-vfm) applies isotropic Gaussian noise to one-hot vectors, meaning the noise magnitude scales with the square root of the number of categories. For high-cardinality features, this likely causes the interpolated signal to be dominated by noise at intermediate timesteps, making it difficult for the model to recover the underlying category identity.
>
> Running the released code under our unified evaluation protocol produces the results below. We include TabFlowM alongside for reference, though we stress that this comparison should be interpreted with the caveats above in mind.
>
> | Dataset | Method | Shape ↓ | Trend ↓ | MLE$^\dagger$ |
> |---|---|---:|---:|---:|
> | Adult | TabbyFlow | 0.0494 | 0.0666 | 0.8916 |
> | Adult | TabFlowM | **0.036** | **0.046** | **0.905** |
> | Beijing | TabbyFlow | 0.3727 | 0.5078 | **0.6759** |
> | Beijing | TabFlowM | **0.040** | **0.043** | 0.739 |
> | Default | TabbyFlow | 0.4321 | 0.5531 | 0.6931 |
> | Default | TabFlowM | **0.057** | **0.042** | **0.747** |
> | Diabetes | TabbyFlow | 0.1271 | 0.4025 | 0.6368 |
> | Diabetes | TabFlowM | **0.057** | **0.104** | **0.651** |
> | Magic | TabbyFlow | 0.0612 | 0.0430 | 0.9055 |
> | Magic | TabFlowM | **0.036** | **0.013** | **0.918** |
> | News | TabbyFlow | 0.1292 | 0.1864 | 0.9428 |
> | News | TabFlowM | **0.067** | **0.021** | **0.837** |
>
> $^\dagger$ MLE is AUC ↑ except for Beijing and News, where it is RMSE ↓.
>
> The results from the released codebase show large degradations on category-heavy datasets such as Default (Shape 0.43, Trend 0.55) and Beijing (Shape 0.37, Trend 0.51), which appears consistent with the noise-scaling issue described above. This pattern makes it difficult to determine whether the performance gap reflects a fundamental methodological limitation of TabbyFlow or an implementation artifact.
>
> The table below summarizes the key architectural differences between the two approaches.
>
> | Design Axis | TabFlowM | TabbyFlow (EF-VFM) |
> |---|---|---|
> | Transport space | Learned latent token space (VAE) | Raw data space |
> | Categorical handling | Unified: categoricals are embedded into continuous tokens; FM operates on a homogeneous space | Per-column: each categorical column gets its own exponential-family likelihood (cross-entropy) |
> | Numerical loss | Standard velocity MSE | Gaussian log-likelihood with time-dependent covariance |
> | Velocity target | Constant: $v^* = \hat{z}_0 - \epsilon$ | Implicit: derived from predicted sufficient statistics |
> | Sampling | Heun (RK2) on a fixed grid | Adaptive ODE solver (dopri5) |
>
> Resolving the implementation ambiguities identified above would require us to make substantive design decisions on behalf of the authors, at which point the comparison would no longer reflect TabbyFlow's method but rather our own interpretation of it. We therefore defer definitive empirical comparison until these specification gaps are clarified by the authors, and instead provide a detailed conceptual comparison in the revised Related Work section highlighting the key architectural differences summarized above: TabbyFlow operates in raw data space with per-column exponential-family likelihoods and an adaptive solver, whereas TabFlowM operates in a unified decoder-compatible latent space with a single global FM objective and fixed-grid Heun integration, trading per-feature distributional flexibility for architectural simplicity and training efficiency.

---

> ### Author Response · Authors · 2026-05-04
> **Response to Reviewer 3UN5 (Concerns 7)**
>
> ### 6. Related Work
>
> ### The Related Work section has been updated to include the requested discussion. Most figures have also been reuploaded to vector format.
>
> ---
>
> ###  Summary
>
> We have also substaintially revised the manuscript: NFE-matched comparisons with trade-off curves including Z-space (Section 7.2); genuine component ablations isolating the coupling path, normalization, token dimension, and adaptive $\beta$ schedule (Sections 7.1--7.3); a strictly controlled Z-space versus token-space comparison under identical VAE architecture (Section 7.1); AUC-PR evaluation across all six BAF configurations (Section 6.2). We believe these additions substantively address the evidence gaps identified by the reviewer.

---

### Review · Reviewer_XmnB · 2026-04-24

**Summary Of Contributions:**

The paper proposes a generative model (coined "TabFlowM") for mixed-type tabular data in the sense that some columns are numerical and some columns are categorical.

The proposed method first tokenizes each column (affine embeddings for numerical columns and embedding tables for categorical columns), and then uses a Transformer-based VAE to map the data to a latent space, and the resulting points are then modelled using a flow model.

The paper provides two propositions shining light on some aspects of the method and the paper concludes with a set of numerical experiments, which seems to favour the proposed method (although I am not familiar with the baseline methods, so I cannot assess the strength of the baselines.)

**Additional Comments:**

Minor:
- Is the sentence missing a word: "... we avoid over-regularizing the VAE latent and ...", e.g. latent variable or similar?

**Audience:**

Yes

**Audience Explanation:**

Although this work may be incremental in nature, there is a very broad interested in generative modelling and flow-based model. Moreover, mixed-type tabular data is of interest to a wide audience, both researchers and practitioners.

**Broader Impact Concerns:**

None.

**Claims And Evidence:**

Yes

**Claims Explanation:**

While the clarity of the paper is lacking (in my opinion) in several places, I believe the main claims are mostly supported. Partially by theoretical arguments (prop. 1 & 2) and a series of numerical experiments in section 5 & 6.

The results of the numerical experiments favours the proposed method, but
1) I am not very familiar with the baseline methods, so it is hard for me to asses the strength of these.
2) It is a bit unclear to me how much "work" is done by the Transformer VAE and the Flow model.

**Requested Changes:**

- The paper uses the phrase "decoder compatible" spaces no less than 20 times, but the paper does not provide a precise definition of what they actually mean by that terminology. Please clarify this.
- Proposition 1 is formulated in somewhat vague language and when first I read the main paper I found it hard to figure out what exactly proposition 1 means. However, after looking at the proof in the appendix, it is more clear. The first part is somewhat trivial (a convex combination of two one-hot encoder vectors is not one-hot encoded vector), and the second part is provides a concrete result assuming regularity conditions on the Hessian and Jacobian of the reconstruction map. In my opinion, it would be much more clear and informative to state this result directly in the text.
- The paper contains lot of vague, but seemingly unjustified statements like e.g.:
	- "The autoencoder stage induces a continuous decoder compatible token representation whose geometry is substantially more regular than that of the original table."
	- "We do not claim the linear path is optimal; rather, its viability hinges on the latentization step, which empirically produces a sufficiently regular token space where such simple Euclidean transport is geometrically plausible."
	- What does these statements mean specificially and what are evidence are they based on?
- In proposition 1, how realistic are the assumptions? The authors should discuss this.
- The overall clarity of the paper could further be improved by including pseudo-code for the training and sample phases.
- The proposed method has many hyperperameters. For example, the reconstruction loss seems to have a hyperparameter for each column and beta-variable for the VAE has \beta_min,  \beta_max, \gamma, \tau. How do the authors deal with those? How sensitive are the results to the choice of these parameters? Please discuss this.
- How necessary is the flow-part? What would the results look like for a well-tuned transformer VAE alone?

---

> ### Author Response · Authors · 2026-05-04
> **Response to Reviewer XmnB (Concerns 1-3)**
>
> We thank the reviewer for the detailed and constructive feedback.
> The revised manuscript addresses most points raised; we summarise
> the key changes below.
>
> ### 1. Precise definition of "decoder-compatible" space
>
> We agree that the original submission used the term without a
> self-contained definition. The revision now includes
> **Definition 1** (Section 4.1), which specifies the two
> properties that a continuous token representation
> $\hat{E}=Dec_\psi(z_{1:})\in R^{M\times d_e}$
> must satisfy to be called *decoder compatible*:
>
> 1. **Column-local decodability.** Each reconstructed value
>    $\hat{x}_m = R_m(\hat{t}_m)$ depends only on the token
>    $\hat{t}_m$ of column $m$, through an affine map (numerical
>    columns) or a linear projection to category logits (categorical
>    columns). Decoding does not couple tokens across columns.
>
> 2. **Reconstruction sufficiency.** Applying these
>    column-local heads to $\hat{E}$ achieves low reconstruction
>    error under the training distribution. That is, column-local
>    decoding is not merely well-defined but sufficient for accurate
>    recovery.
>
> Every subsequent use of the phrase now refers back to this
> definition.
>
> ---
>
> ### 2. Improved statement of Proposition 1
>
> We appreciate the observation that the appendix proof was clearer
> than the main-text statement. In the revision, Proposition 1 is
> now stated in two explicit parts that mirror the proof structure:
>
> > **(Raw-space mismatch).** For any
> > $t \in (0,1)$, the convex combination of two distinct one-hot
> > categorical vectors is not a valid categorical state, so any
> > straight transport path in raw mixed-type space leaves the discrete
> > data manifold and forces the generative model to learn
> > discontinuous corrections.
> >
> > **(Token-space regularity).** If the
> > reconstruction map is twice continuously differentiable with
> > uniformly bounded Hessian and strictly positive minimum Jacobian
> > singular value, then a straight segment in decoder-compatible token
> > space reconstructs to a data-space trajectory whose normalised
> > bending is bounded by a constant that depends only on the
> > decoder's smoothness and conditioning, not on the choice of
> > endpoints.
>
> This formulation makes the first
> claim explicit, while surfacing the concrete regularity conditions
> and the bending bound of the second claim directly in the main
> text.
>
> ---
>
> ### 3. Vague geometric statements replaced with precise references
>
> The two statements the reviewer flagged have been reworded.
> The claim about the autoencoder inducing a "substantially more
> regular" geometry now cites the token space regularity result of Proposition 1 directly and
> explains that the bending bound on decoded trajectories is the
> formal sense in which the token space is more regular than raw
> table space. Similarly, the statement about the viability of the linear path in the introduction to Section 4.2, where we state that "a linear coupling is the simplest admissible interpolant, but is sufficient", now, also references Proposition 1 and its bounding constant to justify why decoder smoothness and non-degeneracy keep the decoded trajectories nearly straight, making the minimal linear coupling adequate.

---

> ### Author Response · Authors · 2026-05-04
> **Response to Reviewer XmnB (Concerns 4)**
>
> ### 4. Realism of the regularity assumptions in Proposition 1
>
> The assumptions of Proposition 1 — bounded Hessian
> $||H_R||_{op} <= \beta$ and strictly positive minimum
> Jacobian singular value $\sigma_{min}(J_R) > 0$ — are best examined
> per reconstruction head (Eq. 6), since Definition 1 ensures that each
> column is decoded independently through its own linear head.
>
> **Numerical columns.**
>
> The reconstruction map for numerical column $i$ is (Eq. 6):
> $\widehat{x}^{num}_i=\widehat{e}^{num}_i\widehat{w}^{num}_i+\widehat{b}^{num}_i$,
> which is affine in the decoded token
> $\widehat{e}^{num}_i \in R^{d_e}$.
>
> - Because the map is affine, its Hessian vanishes identically,
>   so the bounded-Hessian assumption is trivially satisfied
>   with $\beta = 0$ (not to be confused with the KL weight).
>
> - The minimum-singular-value condition reduces to requiring that
>   the learned weight vector
>   $\widehat{w}^{num}_i \in R^{d_e \times 1}$
>   is nonzero (i.e. has positive norm).
>
> - This is not guaranteed architecturally, but is overwhelmingly
>   likely under gradient-based training: a zero weight vector
>   would produce a constant output for all inputs, incurring a
>   large $\ell_{recon}$ penalty (Eq. 8).
>
> **Categorical columns — bounded Hessian.**
>
> The reconstruction map for categorical column $j$ is (Eq. 6):
> $\widehat{p}^{cat}_j=Softmax(\widehat{e}^{cat}_j\widehat{W}^{cat}_j+\widehat{b}^{cat}_j)$,
> which composes a linear projection with a softmax nonlinearity.
>
> - The softmax function is $C^{\infty}$ everywhere, so the
>   Hessian of the composed map is continuous and bounded on any
>   bounded domain.
>
> - Concretely, softmax outputs lie in $[0,1]$ and the Hessian
>   involves only products of these bounded quantities, so the
>   bounded-curvature requirement
>   $||H_R||_{op} <= \beta$ is satisfied for a
>   finite $\beta$.
>
> **Categorical columns — minimum singular value.**
>
> The condition $\sigma_{min}(J_R) > 0$ for categorical heads
> requires more care.
>
> - The softmax maps into the $(C_j - 1)$-simplex
>   $\Delta^{C_j - 1}$, whose outputs sum to one. Its Jacobian
>   $diag(p) - pp^T$ therefore annihilates the
>   all-ones direction $\mathbf{1}$, making it rank-deficient in
>   the ambient $R^{C_j}$ output space.
>
> - Strictly speaking, $\sigma_{min} > 0$ therefore holds only
>   when the output space is understood as the tangent space of
>   the simplex $\Delta^{C_j - 1}$ (dimension $C_j - 1$) rather
>   than full $R^{C_j}$. This is the natural viewpoint
>   in our setting, since generated categorical probabilities are
>   always projected onto the simplex before $argmax$ decoding
>   (Eq. 6).
>
> - Under this intrinsic-dimension view, the condition requires
>   that the Jacobian
>   $(diag(p) - pp^T)\widehat{W}^{cat}_j$
>   has rank $C_j - 1$, i.e. the linear projection
>   $\widehat{W}^{cat}_j \in R^{d_e \times C_j}$
>   does not collapse directions that the softmax Jacobian can
>   still distinguish.
>
> - This is likely under reconstruction-loss training (Eq. 8),
>   but could in principle fail for near-degenerate categories
>   that collapse to a simplex vertex. At a vertex
>   ($p \approx e_k$), the softmax Jacobian
>   $diag(p) - pp^T$ approaches zero, so its
>   non-trivial singular values shrink exponentially with the
>   logit gap, regardless of
>   $\widehat{W}^{cat}_j$.
>
> In our experiments we empirically also do not observe such degeneracies on any of the
> twelve evaluation settings.
>
> - On the six UCI benchmarks, TabFlowM achieves strong column-wise MLE scores
>   (Table 3) and Shape fidelity (Table 1), indicating that the
>   learned reconstruction heads maintain sufficient rank along
>   the transport paths actually traversed by the flow.
>
> - On the six million-scale BAF fraud variants (Table 7), where
>   class imbalance exceeds $100:1$, TabFlowM achieves the
>   best AUC-PR on 5 out of 6 variants and the best Trend on all
>   6. Because this metric directly measures minority-class
>   predictive structure, strong AUC-PR would be impossible if the
>   categorical heads had collapsed to simplex vertices for the
>   rare fraud label: a degenerate head would map all generated
>   tokens to the majority class, destroying the minority signal
>   that AUC-PR measures. The consistently high AUC-PR therefore
>   provides direct evidence that
>   $\sigma_{min}(J_R) > 0$ is maintained even for severely
>   under-represented categories at scale.
>
> **Summary.**
>
> We regard the regularity conditions of Proposition 1 as practically
> satisfied in the regimes explored. The conditions are trivially met for
> numerical heads (affine map, zero Hessian) and naturally satisfied for
> categorical heads under the intrinsic simplex geometry, with the caveat
> that extreme categorical sparsity could in principle violate
> $\sigma_{min} > 0$ near simplex vertices. We flag this as a direction
> for future theoretical refinement.

---

> ### Author Response · Authors · 2026-05-04
> **Response to Reviewer XmnB (Concerns 5-6)**
>
> ### 5. Pseudo-code for training and sampling
>
> The revision includes Algorithm 1 (Training) and Algorithm 2
> (Sampling) at the end of Section 4. For convenience we
> reproduce condensed versions here.
>
> **Training (two stages):**
>
> 1. *Stage 1 (VAE).* For each mini-batch: embed each
>    column into a $d_e$-dimensional token, concatenate into a token
>    matrix $E$; encode to posterior parameters
>    $(\mu, \sigma)$; sample
>    $z = \mu + \sigma \odot \xi$,
>    $\xi \sim N(0, I)$; decode
>    $\hat{E} = Dec_\psi(z)$ and reconstruct each column
>    via its linear head; update encoder and decoder on
>    $\ell_{recon} + \beta KL$ with the adaptive
>    $\beta$ schedule.
>
> 2. *Stage 2 (Flow matching).* Extract the posterior mean
>    $\mu$ (no reparameterisation) for every training row, pass
>    through the frozen decoder to obtain decoder-compatible tokens
>    $\hat{E}$, flatten and normalise to
>    $\tilde z_0 = (z_0 - \bar z)/s$. For each mini-batch: sample
>    $t \sim Unif[0,1]$ and
>    $\varepsilon \sim N(0,I)$; form the interpolant
>    $x_t = (1-t)\varepsilon + t\tilde z_0$; update the velocity
>    network to minimise
>    $||v_\theta(x_t, t) - (\tilde z_0 - \varepsilon)||^2$.
>
> **Sampling:**
>
> Draw $x_0 \sim N(0, I)$; integrate $S$ Heun steps
> using $v_\theta$; de-normalise
> $z^\star = s x_S + \bar{z}$; reshape into an
> $M \times d_e$ token matrix; decode each token through its
> per-column linear head; apply inverse data transforms to obtain
> the synthetic row.
>
> ---
>
> ### 6. Better Ablation/ Hyperparameter sensitivity
>
> The revision substantially expands the ablation study
> (Section 7) as requested by the other reviewers too. We summarize some of the key findings here:
>
> **Coupling path geometry and transport space**
> (Section 7.1, Table 8). We compare the default linear path
> against **quadratic** ($t^2$ schedule) and **cosine**
> variants, as well as **TabFlowM Z-space**, which operates
> on the encoder latent $z$ rather than on decoder-compatible
> tokens. The linear path remains the most balanced choice: the
> quadratic and cosine variants achieve broadly comparable Shape
> and MLE but weaken Trend, likely because their time-varying
> interpolation coefficients introduce non-uniform step sizes
> that degrade pairwise structure. The Z-space variant is
> consistently weaker in both fidelity metrics across both tested
> datasets (Adult and Magic), supporting the choice of
> decoder-compatible token space as the primary transport
> interface.
>
> **Token dimension** (Table 8). We sweep
> $d_e \in \{2, 4, 8, 16\}$ (default $d_e = 4$). Very small
> tokens ($d_e = 2$) severely degrade all metrics. Larger tokens
> ($d_e = 8, 16$) recover Shape and MLE close to the default
> while slightly weakening Trend, indicating moderate sensitivity
> around a sensible middle range.
>
> **Adaptive $\beta$ schedule**
> (Section 7.3, Table 10). We compare the adaptive schedule
> against fixed annealing targets
> $\beta \in \{10^{-2}, 10^{-3}, 10^{-4}, 10^{-5}\}$ on
> BAF-Base. The adaptive schedule achieves the best Trend (0.2090)
> and is essentially tied for the best Shape (0.2306 vs. 0.2303).
> Individual fixed settings can improve one metric but at the cost
> of another: for example, $10^{-3}$ gives the best Shape but
> substantially worsens Trend (0.3625). The adaptive schedule
> therefore offers the most balanced trade-off without requiring
> a per-dataset hyperparameter search.
>
> **NFE sweep** (Section 7.2, Table 9). We sweep the number
> of function evaluations in $\{4, 10, 20, 40, 100\}$ across News,
> Adult, and BAF-Base. TabFlowM achieves the strongest Quality at
> every budget on Adult (reaching 0.041 at NFE = 100 vs. 0.044
> for TabSyn and 0.054 for TabDiff) and converges to near-peak
> quality with substantially fewer evaluations than the diffusion
> baselines, confirming that the efficiency gain is intrinsic to
> the flow matching formulation rather than an artefact of solver
> tuning.

---

> ### Author Response · Authors · 2026-05-04
> **Response to Reviewer XmnB (Concerns 7)**
>
> ### 7. Necessity of the flow-matching stage (VAE-only baseline)
>
> This is an important question. The VAE in TabFlowM serves
> exclusively as a representation learner; it contains no generation
> pathway (i.e. no prior sampling or decoding-from-scratch logic)
> and therefore does not constitute a standalone synthesiser.
> Building a complete transformer-VAE-only pipeline, including
> prior sampling, decoding, inverse data transforms, and
> hyperparameter tuning for generation quality rather than
> reconstruction quality, and running it across all datasets would
> require substantial additional engineering beyond the scope of a
> revision. However, we can triangulate the expected performance of
> such a system from two lines of evidence already available.
>
> 1. **TVAE as a lower bound.**
>    TVAE, an MLP-based VAE synthesiser that generates by sampling
>    $z \sim N(0, I)$ and decoding, is already included as a
>    baseline across all six datasets and consistently underperforms
>    TabFlowM by a wide margin (Tables 1--4). This establishes
>    that VAE-only generation, even with a well-tuned architecture, is
>    insufficient to match methods that learn the latent distribution
>    explicitly.
>
> 2. **The adaptive $\beta$ schedule makes
>    VAE-only generation worse by design.**
>    If one were to build a VAE-only pipeline from our architecture,
>    the generation path would be
>    $z \sim N(0, I_\ell) -> Dec_\psi(z) -> \hat{x}$. This requires the aggregate posterior
>    $q(z) = (1/N)\sum_i q_\phi(z | x_i)$ to closely match
>    the standard Gaussian prior, since any mismatch means the model
>    samples from low-density regions of the learned latent space.
>    Our adaptive schedule (Eq. 10), however, deliberately reduces
>    $\beta$ once reconstruction plateaus, prioritising token-level
>    fidelity for the downstream flow model at the direct cost of
>    prior alignment. As a result, the aggregate posterior deviates
>    substantially from $N(0, I_\ell)$, and a VAE-only
>    generator built on these weights would likely perform
>    *worse* than TVAE, which trains with fixed $\beta$ and
>    maintains tighter prior alignment. The low-$\beta$ regime is not
>    a limitation; it is a deliberate design choice that trades prior
>    match for reconstruction quality, with the flow matching stage
>    compensating by learning the actual token-space distribution.
>
> Taken together, these observations strongly suggest that the
> flow-matching stage is not merely a marginal addition but is
> essential to TabFlowM's performance. The VAE constructs a
> well-behaved transport space; the flow model learns the
> distribution within that space. Removing either component would
> forfeit the corresponding benefit.
>
> We hope these revisions and clarifications address
> the reviewer's concerns. We are happy to discuss any remaining
> points further.

---

### Decision · Action_Editor_LCGk · 2026-06-20

**Recommendation:** Accept with minor revision

**Additional Comments:**

All three reviewers recommend Leaning Accept after the revision, and the authors' substantial experimental additions (component ablations, NFE sweeps, Z-space comparison, AUC-PR evaluation, rare-category analysis) have addressed the major evidentiary gaps identified in the initial reviews. The paper is technically sound and empirically thorough in its revised form, making it suitable for TMLR publication. I request the following minor revisions for the camera-ready version:

1. Clarify the remaining concern from Reviewer UN33, who noted that "there still have concerns need to be solve" in the final recommendation without further specification. If these refer to the original issues already addressed by the revision, state this explicitly. If additional experiments or clarifications are needed, incorporate them.

2. Ensure the formal Definition 1 ("decoder-compatible") and the revised Proposition 1 (with explicit regularity conditions and the two-part structure) are fully integrated into the main text as committed in the rebuttal. The discussion of when the regularity assumptions may fail (e.g., near simplex vertices for extremely sparse categories) should appear alongside the proposition, not only in the rebuttal.

3. Include the pseudo-code for training and sampling (Algorithms 1 and 2) as committed.

4. The comparison with TabbyFlow should be presented with appropriate caveats about the implementation ambiguities encountered, as outlined in the rebuttal, so readers can draw their own conclusions.

5. Ensure all figures are in vector format as committed, and that the corrected table entries (BAF-V3 TabDiff Shape, News cflow MLE) are finalized.

6. The abstract and claims throughout should consistently refer to training-time (not sampling-time) efficiency advantages, and the composite fidelity framing should avoid treating Shape and Trend as fully independent metrics.

**Audience:**

Yes

**Audience Explanation:**

All three reviewers affirmed audience interest. Mixed-type tabular data synthesis is a practically important area with broad relevance to both researchers and practitioners. The community is actively exploring efficient alternatives to computationally heavy diffusion models, and the paper's demonstration that a simple flow matching objective in an appropriately constructed latent space can match or exceed diffusion-based methods while reducing training time and conceptual complexity provides a useful reference point. The large-scale BAF fraud dataset experiments, addressing extreme class imbalance at million-row scale, target a genuine industrial pain point. Even reviewers who viewed the contribution as incremental acknowledged its value as a well-characterized baseline for the field.

**Claims And Evidence:**

Yes

**Claims Explanation:**

The paper proposes TabFlowM, a two-stage framework that first trains a Transformer-based VAE to map mixed-type tabular records into a continuous, decoder-compatible token space, and then learns a velocity field via flow matching with a linear coupling path to transport Gaussian noise to the learned data distribution. The central claim is that this minimalist flow matching formulation can recover most of the benefits of heavier diffusion-based tabular synthesizers (TabSyn, TabDiff) while substantially reducing computational and conceptual complexity.

The initial reviews identified significant evidentiary gaps: the ablation study conflated backbone replacement with component isolation (Reviewers 3UN5 and UN33), NFE (number of function evaluations) details were missing (Reviewer 3UN5), the BAF evaluation relied on ROC-AUC rather than the more appropriate AUC-PR under extreme class imbalance (Reviewer 3UN5), and several vague statements and undefined terminology weakened precision (Reviewer XmnB). The authors undertook a substantial revision that addressed these concerns comprehensively. The revised manuscript includes genuine component ablations isolating the coupling path geometry (linear vs. quadratic vs. cosine), the transport space (decoder-compatible tokens vs. encoder z-space), token normalization, token dimension, and the adaptive β schedule — all under identical VAE weights, velocity network, and solver. The NFE sweep (4 to 100 evaluations) demonstrates that TabFlowM's quality advantage is intrinsic to the flow matching formulation and not an artifact of solver budget. The Z-space comparison provides controlled evidence that the decoder-compatible token space yields a lower quality floor at convergence. The BAF evaluation was revised to use AUC-PR as the primary metric, with TabFlowM achieving the strongest average AUC-PR across five of six variants. The argmax-vs-softmax decoding analysis and per-category frequency statistics further support rare-category preservation. Table typos were corrected and the abstract's erroneous sampling-time claim was fixed to refer to training time. Proposition 1 was restated with explicit regularity conditions, and "decoder-compatible" received a formal definition.

Some minor residual concerns remain: Reviewer UN33 noted that some concerns "still need to be solved" in the final recommendation but did not specify which, and all three reviewers characterized the contribution as a well-executed integration of existing paradigms (latent tabular generation + flow matching) rather than a fundamentally novel algorithmic advance. Nonetheless, the empirical evidence is now thorough, the claims are appropriately scoped, and all three reviewers converged on Leaning Accept.